# Compensation for PKMζ in long-term potentiation and spatial long-term memory in mutant mice

Panayiotis Tsokas[1,2], Changchi Hsieh[1], Yudong Yao[1], Edith Lesburguères[3], Emma Jane Claire Wallace[1], Andrew Tcherepanov[1], Desingarao Jothianandan[1], Benjamin Rush Hartley[1], Ling Pan[1], Bruno Rivard[1], Robert V Farese[4], Mini P Sajan[4], Peter John Bergold[1], Alejandro Iván Hernández[5], James E Cottrell[2], Harel Z Shouval[6], André Antonio Fenton[1,3]*, Todd Charlton Sacktor[1,2,7]*

[1]Department of Physiology and Pharmacology, The Robert F Furchgott Center for Neural and Behavioral Science, State University of New York Downstate Medical Center, Brooklyn, United States; [2]Department of Anesthesiology, State University of New York Downstate Medical Center, Brooklyn, United States; [3]Center for Neural Science, New York University, New York, United States; [4]Department of Internal Medicine, James A Haley Veterans Hospital, University of South Florida, Tampa, United States; [5]Department of Pathology, State University of New York Downstate Medical Center, Brooklyn, United States; [6]Department of Neurobiology and Anatomy, University of Texas Medical School at Houston, Houston, United States; [7]Department of Neurology, State University of New York Downstate Medical Center, Brooklyn, United States

**\*For correspondence:** afenton@ nyu.edu (AAF); tsacktor@ downstate.edu (TCS)

**Competing interests:** The authors declare that no competing interests exist.

**Abstract** PKMζ is a persistently active PKC isoform proposed to maintain late-LTP and long-term memory. But late-LTP and memory are maintained without PKMζ in PKMζ-null mice. Two hypotheses can account for these findings. First, PKMζ is unimportant for LTP or memory. Second, PKMζ is essential for late-LTP and long-term memory in wild-type mice, and PKMζ-null mice recruit compensatory mechanisms. We find that whereas PKMζ persistently increases in LTP maintenance in wild-type mice, PKCι/λ, a gene-product closely related to PKMζ, persistently increases in LTP maintenance in PKMζ-null mice. Using a pharmacogenetic approach, we find PKMζ-antisense in hippocampus blocks late-LTP and spatial long-term memory in wild-type mice, but not in PKMζ-null mice without the target mRNA. Conversely, a PKCι/λ-antagonist disrupts late-LTP and spatial memory in PKMζ-null mice but not in wild-type mice. Thus, whereas PKMζ is essential for wild-type LTP and long-term memory, persistent PKCι/λ activation compensates for PKMζ loss in PKMζ-null mice.

## Introduction

LTP and long-term memory can be divided into two mechanistically distinct phases—a transient induction and a persistent maintenance (*Malinow et al., 1988*). Induction is thought to rely solely on post-translational modifications. Maintenance requires new protein synthesis soon after strong synaptic stimulation or learning, and these newly synthesized proteins are believed to sustain the synaptic potentiation or behavioral modification (*Kandel and Schwartz, 1982*). Although numerous signal transduction molecules are important for LTP and long-term memory, most have been implicated in induction, with many participating in either the initial transient potentiation or the mechanisms for

**eLife digest** How are long-term memories stored in the brain? The formation of memories is believed to depend on the strengthening of connections between neurons. During learning, neurons produce an enzyme called PKMzeta (or PKMζ), which is thought to be responsible for maintaining the newly strengthened connections. Inhibitors of PKMzeta, such as a drug called ZIP, disrupt long-term memories. This suggests that the brain may be like a computer hard disc in that its stored information — its memories — could be erased.

However, recent experiments on genetically engineered mice have thrown the role of PKMzeta into question. Knockout mice that lack the gene for PKMzeta can still strengthen connections between neurons and can still learn and remember. Moreover, ZIP still works to reverse the strengthening and to erase long-term memories. This indicates that ZIP can act on something other than the PKMzeta enzyme. These results have led many neuroscientists to doubt that PKMzeta has anything to do with memory.

Yet there are two possible explanations for the normal memory in PKMzeta knockout mice. First, PKMzeta is not required for memory, so getting rid of it has no effect. Second, PKMzeta is essential for long-term memory in *normal* mice. However, knockout mice recruit a back-up mechanism for long-term memory storage, which is also sensitive to the effects of ZIP.

To test these possibilities, Tsokas et al. used a modified piece of DNA that prevents neurons with the gene for PKMzeta from producing the enzyme. The DNA blocked memory formation in normal mice, consistent with a role for PKMzeta in memory. However, it had no effect in knockout mice — the DNA had nothing to work on. This suggests that another molecule does indeed act as a back-up for PKMzeta in these animals. Further experiments revealed that an enzyme closely related to PKMzeta, called PKCiota/lambda (PKCι/λ), substitutes for PKMzeta during memory storage in the knockout mice.

These findings restore PKMzeta to its early promise. They show that PKMzeta is crucial for long-term memory in normal mice, but that something as important as memory storage has a back-up mechanism should PKMzeta fail. Future work may reveal when and how this back-up becomes engaged.

upregulating new protein synthesis (*Sanes and Lichtman, 1999*). In contrast, few molecules have been implicated in maintenance. Such a maintenance molecule would be both: 1) a product of de novo protein synthesis sufficient to enhance synaptic transmission and 2) a necessary component of the mechanism sustaining synaptic potentiation and long-term memory, as shown by inhibition of the molecule reversing sustained synaptic potentiation and erasing long-term memory.

One hypothesis for a molecular mechanism of maintenance involves the persistent increase in an autonomously active PKC isoform, PKMζ (*Sacktor, 2011*). LTP and long-term memory maintenance may depend upon the function of PKMζ because data suggest the kinase possesses the two essential properties of a maintenance molecule. First, PKMζ is produced in LTP by new protein synthesis and is sufficient to potentiate synaptic transmission. PKMζ is generated from a PKMζ mRNA, but this mRNA is translationally repressed in the basal state of neurons (*Hernandez et al., 2003*). During LTP, strong afferent synaptic stimulation derepresses the mRNA and rapidly increases the de novo synthesis of PKMζ (*Hernandez et al., 2003*). The newly synthesized kinase, unlike most other protein kinases, is autonomously active without the requirement for second messenger stimulation (*Sacktor et al., 1993*; *Hernandez et al., 2003*), and the autonomous activity of PKMζ is sufficient to enhance synaptic transmission (*Ling et al., 2002*; *2006*; *Yao et al., 2008*). Second, multiple inhibitors of PKMζ and a dominant-negative mutated form of PKMζ reverse established LTP maintenance or disrupt long-term memory storage (*Ling et al., 2002*; *Serrano et al., 2005*; *Pastalkova et al., 2006*; *Shema et al., 2011*; *Cai et al., 2011*).

Recently, this proposed function of PKMζ has been challenged by new findings that LTP and long-term memory appear normal in PKMζ-null mice (*Lee et al., 2013*; *Volk et al., 2013*). Moreover, the PKMζ-inhibitor ZIP, which disrupts the maintenance of LTP and long-term memory in wild-type animals, disrupts these same processes in PKMζ-null mice (*Lee et al., 2013*; *Volk et al., 2013*). Two

hypotheses can account for these findings (*Frankland and Josselyn, 2013*; *Matt and Hell, 2013*). First, in a straightforward hypothesis, PKMζ is unnecessary for LTP or long-term memory, and therefore genetically deleting PKMζ has no effect on these processes (*Lee et al., 2013*; *Volk et al., 2013*). Second, PKMζ is essential for late-LTP and long-term memory in wild-type mice, and compensatory mechanisms emerge in the mutant mice to substitute for PKMζ, which are also inhibited by ZIP. Here, we used a pharmacogenetic approach to distinguish between the 'PKMζ is unnecessary hypothesis' and the 'PKMζ is compensated hypothesis'.

## Results

### ZIP reverses late-LTP maintenance in both wild-type and PKMζ-null mice and blocks synaptic potentiation produced by PKMζ and PKCι/λ

We first confirmed the previously published findings that late-LTP appears similar in PKMζ-null and wild-type mice, and that ZIP (5 μM) applied to the bath 3 hr after tetanization reverses late-LTP in both wild-type mice (*Serrano et al., 2005*) and PKMζ-null mice (*Volk et al., 2013*, *Figure 1A,B*). In interface chambers in which maximal drug concentrations are achieved slowly, the reversal of late-LTP may be more rapid in wild-type mice than in PKMζ-null mice (time to 50% of the pre-ZIP response is different: wild-type, $129 \pm 28$ min; PKMζ-null, $311 \pm 72$ min; $t_7 = 2.57$, p = 0.037, d = 1.64).

Because these results clearly show that ZIP affects molecules that can maintain synaptic potentiation in addition to PKMζ, we examined whether ZIP blocks synaptic enhancement produced by other PKC isoforms. PKC is a gene family, consisting of conventional, novel, and atypical isoform classes (*Nishizuka, 1988*). ZIP inhibits the phosphotransferase activity of both PKMζ and, at higher doses, the other atypical PKC, PKCι/λ (mouse PKCλ and human PKCι are orthologous genes) (*Ren et al., 2013*) (*Figure 1C*). We found that ZIP at the concentration standardly used to reverse late-LTP (5 μM) (*Serrano et al., 2005*) blocks the synaptic potentiation produced by postsynaptic perfusion of both atypical PKCs (*Figure 1D,E*). In contrast, this dose of ZIP does not block the synaptic potentiation produced by conventional/novel PKCs activated by phorbol esters (*Figure 1—figure supplement 1A*). This dose of ZIP also has no effect on basal properties of neurons, such as field excitatory postsynaptic potentials (fEPSPs) in non-tetanized synaptic pathways recorded within the slices of the wild-type and PKMζ-null mice (*Figure 1A,B*) or on the membrane stability of CA1 pyramidal cells (*Figure 1—figure supplement 1B*).

### PKMζ-antisense prevents late-LTP in wild-type mice, but not PKMζ-null mice

Because ZIP blocks the synaptic potentiation produced by both PKMζ and PKCι/λ, it is unsuitable for evaluating PKMζ's function in pharmacogenetic experiments. We therefore took advantage of the specific nucleotide sequence of the translation start site of the PKMζ mRNA that has no significant homology with sequences in any other known mRNA, except PKCζ mRNA that is not expressed in the hippocampus (*Hernandez et al., 2003*). We hypothesized that if there were compensation for PKMζ's function in PKMζ-null mice during LTP, a PKMζ-antisense oligodeoxynucleotide should prevent late-LTP in wild-type mice, but not in PKMζ-null mice that lack the antisense's target mRNA (*Figure 2A*, *Figure 2—figure supplement 1*).

We first validated that bath applications of PKMζ-antisense to hippocampal slices from wild-type mice block the new synthesis of PKMζ and not other gene products induced during LTP (*Figure 2B*). PKMζ-antisense selectively blocks the increase of PKMζ during LTP, but not the activity-dependent increase of PKCι/λ, which occurs transiently in LTP (*Osten et al., 1996*; *Kelly et al., 2007*), or a third protein that is rapidly synthesized in LTP, eukaryotic elongation factor 1A (eEF1A) (*Tsokas et al., 2005*) (*Figure 2B*). The brief application of the PKMζ-antisense has no effect on basal levels of PKMζ, as expected for the relatively long half-life of the kinase in untetanized hippocampal slices (*Osten et al., 1996*), or on basal levels of PKCι/λ (*Figure 2—figure supplement 2*).

We then tested the effect of PKMζ-antisense on late-LTP in wild-type and PKMζ-null mice. Whereas PKMζ-antisense blocks late-LTP in wild-type mice, the same antisense has no effect on late-LTP in PKMζ-null mice (*Figure 2C*). These results are predicted by the PKMζ is

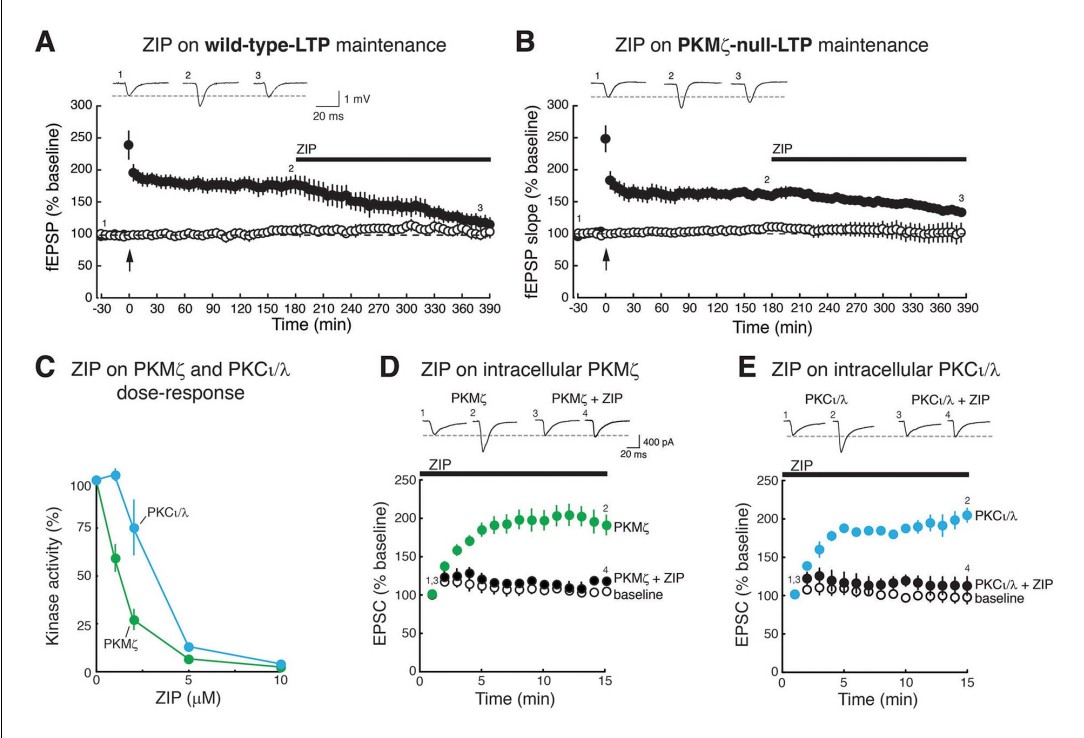

**Figure 1.** ZIP reverses LTP maintenance in both wild-type mice and PKMζ-null mice and blocks synaptic potentiation mediated by PKMζ and PKCι/λ. Bath applications of ZIP (5 µM) reverse (A) wild-type-LTP maintenance and (B) PKMζ-null-LTP maintenance (filled circles). Above insets, numbered representative fEPSP traces correspond to time points noted below. Below, mean ± SEM. For (A), wild-type, n = 5, average response 5 min before ZIP compared to 3.5 hr post-ZIP, $t_4 = 4.83$, p = 0.0084, d = 1.85; for (B), PKMζ-null, n = 4, $t_3 = 3.34$, p = 0.045, d = 2.88. Non-tetanized pathways are stable in the presence of ZIP (open circles). For (A), wild-type non-tetanized pathway: 5 min pre-ZIP vs. 3.5 hr post-ZIP; n = 5, $t_4 = 1.73$, p = 0.16; d = 0.49; for (B), PKMζ-null non-tetanized pathway: n = 4, $t_3 = 0.82$, p = 0.47, d = 0.058. (C) ZIP inhibits both PKMζ and, at higher doses, the autonomous activity of PKCι/λ. The main effects and interactions are all significant (kinase: $F_{1,30} = 85.4$, p = $2.77 \times 10^{-10}$, $\eta^2 = 0.036$; ZIP concentration: $F_{4,30} = 200.56$, p = $3.48 \times 10^{-21}$, $\eta^2 = 0.34$; interaction: $F_{4,30} = 26.59$, p = $1.98 \times 10^{-9}$, $\eta^2 = 0.045$). Post-hoc tests show the kinases respond differently at 1 µM and 2 µM ZIP. (D, E), ZIP blocks EPSC potentiation produced by postsynaptic dialysis of PKMζ or PKCι/λ in CA1 pyramidal cells in hippocampal slices. ZIP (5 µM) is applied to the bath prior to obtaining whole-cell patch. Above insets, numbered representative EPSC traces correspond to time points noted below. Statistical comparisons are at 15 min after whole-cell patch. (D) PKMζ: n's = 4, $F_{2,11} = 18.07$, p = 0.0007, d = 1.78; post-hoc tests: PKMζ vs. baseline, p = 0.0012; PKMζ vs. PKMζ + ZIP, p = 0.0018; PKMζ + ZIP vs. baseline, p = 0.94. (E) PKCι/λ: n's = 4, $F_{2,11} = 35.2$, p = $1.66 \times 10^{-5}$, d = 1.79; post-hoc tests: PKCι/λ vs. baseline, p<0.0001; PKCι/λ vs. PKCι/λ + ZIP, p<0.0001; PKCι/λ + ZIP vs. baseline, p = 0.86.

The following figure supplements are available for figure 1:

**Figure supplement 1.** ZIP has no effect on synaptic potentiation induced by activation of conventional/novel PKCs and produces no perturbation of neuronal membrane conductance.

**Figure supplement 2.** Chelerythrine inhibits both PKMζ and PKCι/λ.

compensated hypothesis but not by the PKMζ is unnecessary hypothesis. Control scrambled oligodeoxynucleotides have no effect on either wild-type- or PKMζ-null-LTP, further supporting the conclusion that the antisense effect is selective for PKMζ.

## PKCι/λ-inhibition reverses late-LTP maintenance in PKMζ-null mice, but not wild-type mice

What maintains LTP in PKMζ-null mice? We first measured the basal levels of all eight remaining PKC isoforms expressed in the dorsal hippocampus of PKMζ-null mice and found increased basal amounts of PKCι/λ and the conventional PKC, PKCβI (*Figure 3—figure supplement 1,2A*). We focused on PKCι/λ because it is the most closely related gene product to PKMζ and is blocked by ZIP (*Figure 1C,E*). PKCι/λ, which is important for synaptic potentiation during an early phase of LTP

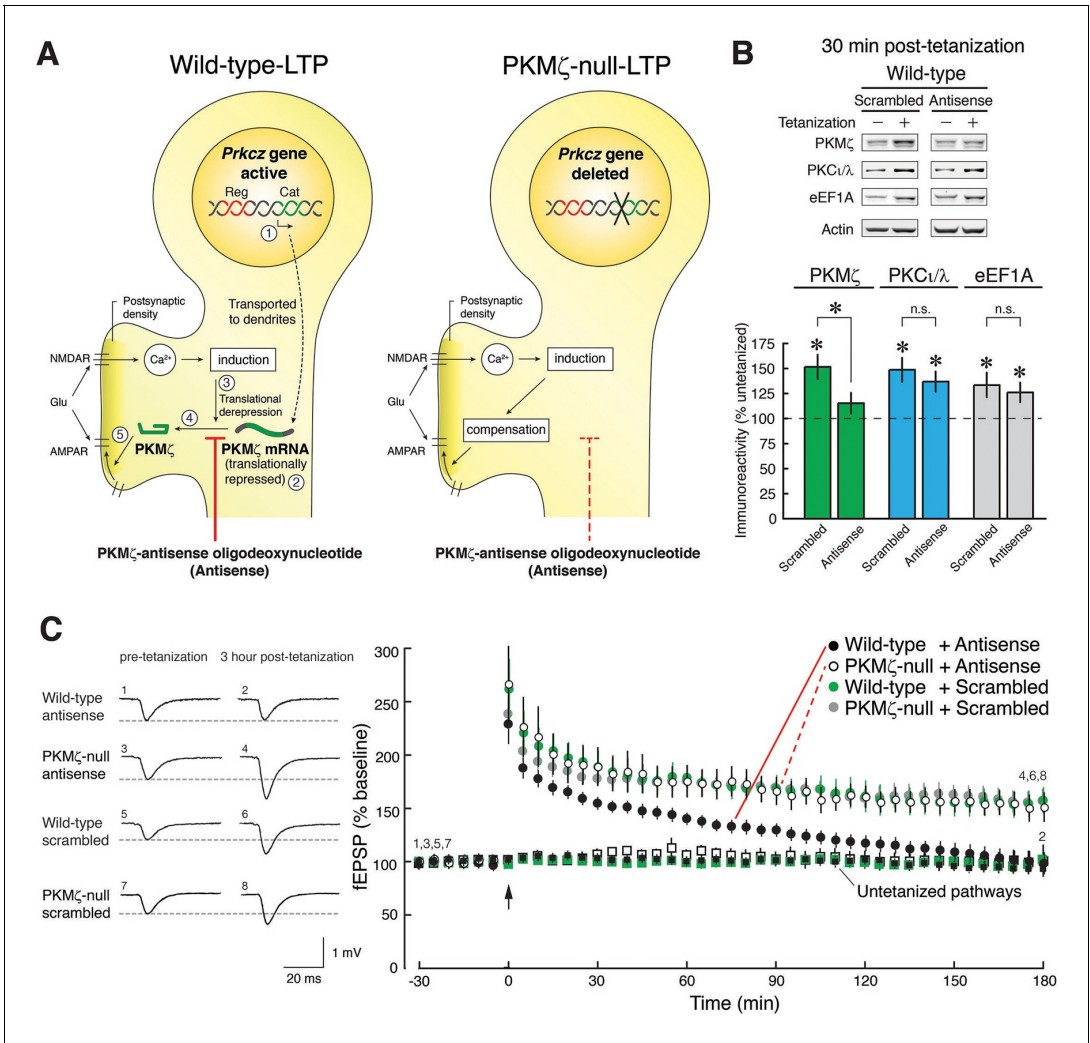

**Figure 2.** PKMζ is essential for late-LTP in wild-type mice, and compensation accounts for late-LTP in PKMζ-null mice. (**A**) Diagram illustrating the PKMζ-compensation hypothesis tested by pharmacogenetic analysis of LTP. The *Prkcz* gene consists of an autoinhibitory PKCζ regulatory domain exon region (Reg, shown in red) and a catalytic domain exon region (Cat, green). In neurons in wild-type mice, PKMζ is produced by an internal promoter within the *Prkcz* gene, transcribing a PKMζ mRNA that expresses an independent ζ catalytic domain (indicated as step [1] [*Hernandez et al., 2003*]). The PKMζ mRNA is transported to dendrites (*Muslimov et al., 2004*) but is translationally repressed ([2] [*Hernandez et al., 2003*]). During tetanic stimulation, glutamate (Glu) activates the NMDAR to stimulate $Ca^{2+}$-dependent induction mechanisms that release the translational block ([3] [*Hernandez et al., 2003*]), resulting in synthesis of PKMζ ([4] [*Hernandez et al., 2003*]), which potentiates postsynaptic AMPARs ([5] [*Ling et al., 2002*; *Serrano et al., 2005*]). If wild-type mice express persistently enhanced AMPAR-mediated synaptic transmission through synthesis of PKMζ and PKMζ-null mice through compensatory mechanisms, then PKMζ-antisense will block LTP in wild-type mice (left) but have no effect in PKMζ-null mice (right). (**B**) The PKMζ-antisense (20 μM) blocks the new synthesis of PKMζ, but not PKCι/λ or the eukaryotic elongation factor 1A (eEF1A) that are also rapidly synthesized in LTP. In the presence of antisense or scrambled oligodeoxynucleotides, adjacent slices from the same hippocampus are either tetanized or untetanized, and 30-min post-tetanization CA1 regions are assayed by immunoblot. The levels of protein in the untetanized slices are set at 100%. Mean ± SEM; *denotes significance; n.s., no significance. PKMζ: scrambled, tetanized (n = 17) vs. untetanized (n = 19), $t_{34}$ = 3.81, p = 0.00056, d = 1.27; antisense, tetanized (n = 12) vs. untetanized (n = 18), $t_{28}$ = 1.35, p = 0.19, d = 0.50; antisense vs. scrambled, $t_{27}$ = 2.12, p = 0.043, d = 0.80; PKCι/λ: scrambled, tetanized (n = 17) vs. untetanized (n = 18), $t_{33}$ = 3.72, p = 0.00074, d = 1.26; antisense, tetanized (n = 12) vs. untetanized (n = 17), $t_{27}$ = 3.59, p = 0.0013, d = 1.35; antisense vs. scrambled, $t_{27}$ = 0.71, p = 0.49, d = 0.27; eEF1A: scrambled, tetanized (n = 9) vs. untetanized (n = 10), $t_{17}$ = 2.40, p = 0.028, d = 1.10; antisense, tetanized (n = 12) vs. untetanized (n = 18), $t_{28}$ = 2.07, p = 0.048, d = 0.77; antisense vs. scrambled, $t_{19}$ = 0.47, p = 0.64, d = 0.21. (**C**) PKMζ-antisense blocks late-LTP in wild-type mice but has no effect on LTP in PKMζ-null mice. Left, representative fEPSPs; numbers correspond to time points at right. Right, mean ± SEM. Scrambled version of the oligodeoxynucleotide has no effect on either genotype. Circles show tetanized pathways, and color-coded squares show control untetanized pathways within each slice that receive only test stimulation. Tetanization is at arrow. N's are: wild-type + antisense: 7; PKMζ-null + antisense: 5; wild-type + scrambled: 8; PKMζ-null + scrambled: 6. The genotype X drug X time ANOVA with repeated measures on time (average of the 5 min ending at 30 min post-tetanization and 180 min post-tetanization) confirmed a significant genotype X

*Figure 2 continued on next page*

*Figure 2 continued*

drug interaction, $F_{1,22}$ = 4.7; p = 0.041, $\eta^2$ = 0.0026. Post-hoc tests confirmed antisense on wild-type at 180 min post-tetanization is significantly less than all other responses.

The following figure supplements are available for figure 2:

**Figure supplement 1.** The mutant PKMζ gene expresses neither PKMζ mRNA nor protein.

**Figure supplement 2.** PKMζ-antisense bath-applied to hippocampal slices for 2 hr does not affect the amount of basal PKMζ or PKCι/λ.

through post-translational activation (*Kelly et al., 2007*; *Ren et al., 2013*), also has partial activity in the absence of lipid second messengers (*Akimoto et al., 1994*) (*Figure 3—figure supplement 2B*) and total levels of PKCι/λ increase transiently after tetanic stimulation in wild-type animals (*Osten et al., 1996*; *Kelly et al., 2007*).

We therefore examined the possibility that the transient increase in PKCι/λ in wild-type-LTP changes in PKMζ-null-LTP (*Figure 3*). As expected, in wild-type mice, the increase of PKCι/λ observed 30 min post-tetanization (*Figure 2B*) returns to basal levels by 3 hr (*Osten et al., 1996*; *Kelly et al., 2007*) (*Figure 3A*). But in PKMζ-null mice, the increase in PKCι/λ during LTP lasts at least 3 hr, persisting like the increase of PKMζ in LTP maintenance in wild-type mice (*Figure 3A*). These data suggest PKCι/λ as a candidate for functionally compensating for the loss of PKMζ in the PKMζ-null mice.

We therefore tested the possible involvement of persistent increased activity of PKCι/λ in maintaining PKMζ-null-LTP. Inhibiting PKCι/λ blocks early-LTP and thus prevents the formation of late-LTP in wild-type animals (*Ren et al., 2013*). Therefore, in order to test PKCι/λ's potential function in maintaining LTP in PKMζ-null mice, we examined the effects of a PKCι/λ-antagonist applied only after late-LTP had been established. We used the cell-permeable antagonist [4-(5-amino-4-carbamoylimidazol-1-yl)-2,3-dihydroxycyclopentyl] methyl dihydrogen (ICAP) that blocks PKCι/λ, and not ζ kinase activity (*Pillai et al., 2011*; *Sajan et al., 2013*; *2014*). We first induced LTP in PKMζ-null and wild-type mice and then applied ICAP to the bath 3 hr later during late-LTP maintenance (*Figure 3B,C*). The PKCι/λ-inhibitor reverses established PKMζ-null-LTP maintenance but has no effect on wild-type-LTP maintenance. To test the inhibitor's efficacy in the wild-type slices in which no effect on late-LTP maintenance can be observed, we tetanized a separate synaptic pathway in the presence of the antagonist and found that it blocks early-LTP induction, as expected (*Ren et al., 2013*) (*Figure 3C*). The selective effect of the PKCι/λ-inhibitor on only PKMζ-null-LTP maintenance is predicted by the PKMζ-compensation hypothesis, but not by the PKMζ-unnecessary hypothesis.

## PKMζ-antisense blocks spatial long-term memory in wild-type mice, but not PKMζ-null mice

We used the pharmacogenetic approach to re-examine PKMζ function in hippocampus-dependent spatial memory. General protein synthesis inhibitors are effective in blocking long-term memory in a time-window around the time of conditioning (*Davis and Squire, 1984*) and intrahippocampally injected antisense lasts at least 2 hr (*Garcia-Osta et al., 2006*). Therefore, for PKMζ-antisense to be present throughout spatial conditioning, we examined active place avoidance, a conditioned behavior that mice can rapidly learn and remember after three 30 min training sessions, spaced 2 hr apart (*Figure 4*). Following intrahippocampal injections of control scrambled oligodeoxynucleotides, 1-day memory retention in PKMζ-null and wild-type mice appears the same. In contrast, PKMζ-antisense disrupts 1-day long-term memory in wild-type mice but not in PKMζ-null mice (*Figure 4*, *Figure 4—figure supplement 1*). These results are predicted by the PKMζ-compensation hypothesis but not the PKMζ-unnecessary hypothesis.

## PKCι/λ-inhibition disrupts spatial long-term memory maintenance in PKMζ-null mice, but not wild-type mice

We then examined whether the persistent action of PKCι/λ maintains spatial long-term memory in PKMζ-null mice. Analogous to the LTP experiments, we tested PKCι/λ's function in spatial long-term

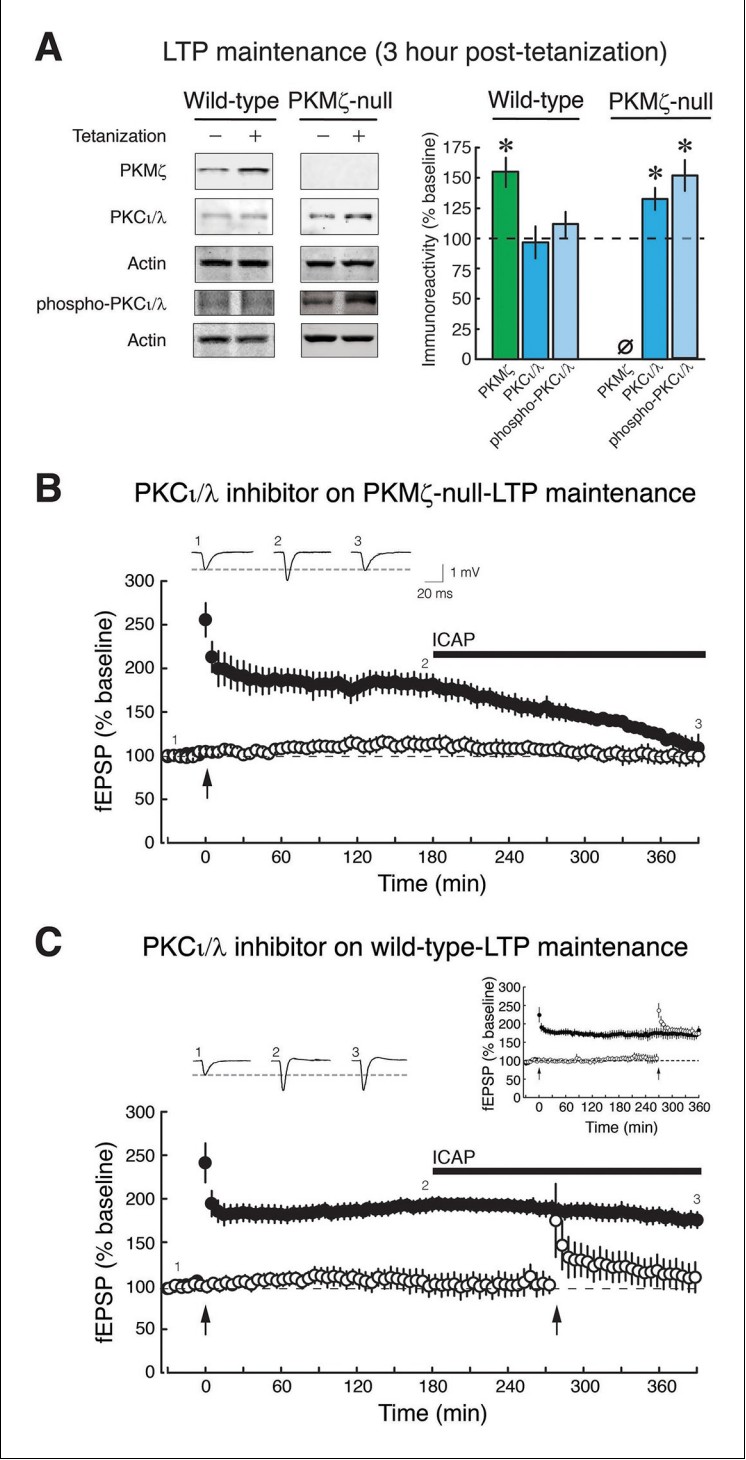

**Figure 3.** PKCι/λ inhibitor reverses the maintenance of late-LTP in PKMζ-null mice, but not in wild-type mice. (**A**) Immunoblots show that in wild-type-LTP maintenance, PKMζ persistently increases for 3 hr, whereas PKCι/λ, as determined by both total and activation loop phosphorylated-PKCι/λ antisera, are at baseline. In PKMζ-null-LTP maintenance, both total and activation loop phosphorylated-PKCι/λ persistently increase for 3 hr. Wild-type: PKMζ, untetanized vs. tetanized (n's = 9), $t_{16}$ = 4.51, p = 0.00036, d = 2.13; PKCι/λ, untetanized (n = 6) vs. tetanized (n = 7), $t_{11}$ = 0.19, p = 0.85, d = 0.11; phospho-PKCι/λ, untetanized vs. tetanized (n's = 9), $t_{16}$ = 0.86, p = 0.40, d = 0.41. PKMζ-null: PKCι/λ, untetanized vs. tetanized (n's = 7), $t_{12}$ = 2.41, p = 0.033, d = 1.29; phospho-PKCι/λ, untetanized (n = 8) vs. tetanized (n = 9), $t_{15}$ = 4.35, p = 0.00058, d = 2.11. PKMζ-null: tetanized total PKCι/λ vs. phospho-PKCι/λ, $t_{14}$ = 1.83, p = 0.09, d = 0.92. Tetanized wild-type vs. PKMζ-null PKCι/λ, $t_{12}$ = 2.28, p =

*Figure 3 continued on next page*

*Figure 3 continued*

0.042, d = 1.22; tetanized wild-type vs. PKMζ-null phospho-PKCι/λ, $t_{16}$ = 3.27, p = 0.0048, d = 1.54. (**B**) PKMζ-null late-LTP maintenance (filled circles) is reversed by PKCι/λ-antagonist ICAP (10 µM) applied 3 hr post-tetanization. Insert above, representative fEPSPs; numbers correspond to time points below. Below, mean ± SEM. Comparing average responses of the 5 min before drug and 3.5 hr after drug, n = 4, $t_3$ = 5.4, p = 0.012, d = 3.22. ICAP has no effect on a second, independent synaptic pathway recorded within the slices that received no tetanization (open circles). (**C**) ICAP has no effect on wild-type LTP maintenance (filled circles; n = 7, $t_6$ = 1.88, p = 0.11, d = 0.67), but blocks the initial potentiation following tetanization (right arrow) in the second synaptic pathway (open circles). The effect of ICAP is different on LTP maintenance in the wild-type and PKMζ-null; $t_9$ = 2.75, p = 0.023, d = 1.129. Right insert above, inhibition of LTP induction is not due to prolonged perfusion in vitro because tetanization of a second pathway recorded for equivalent periods of time induces LTP.

The following figure supplements are available for figure 3:

**Figure supplement 1.** Analysis of the complete PKC isoform family shows increases in basal expression of PKCι/λ and PKCβI in the dorsal hippocampus of PKMζ-null mice.

**Figure supplement 2.** Characterization of PKCι/λ immunoreactivity, molecular weight, and kinase activity.

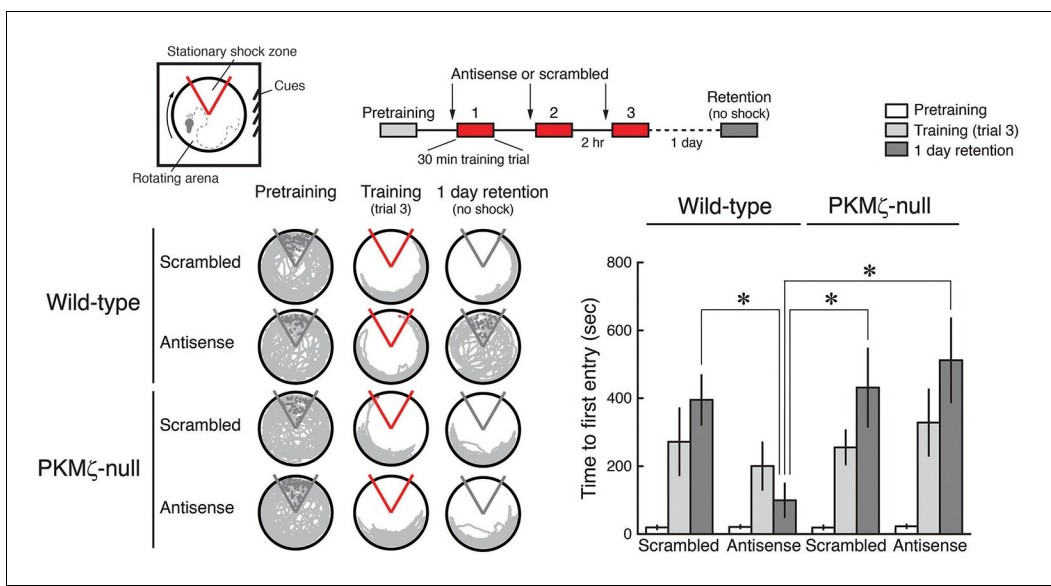

**Figure 4.** PKMζ is essential for spatial long-term memory in wild-type mice, and compensation accounts for spatial long-term memory in PKMζ-null mice. PKMζ-antisense blocks spatial long-term memory in wild-type mice but has no effect on long-term memory in PKMζ-null mice. Inserts above, (left) schematic diagram of active place avoidance training apparatus and (middle) 1-day training protocol. Intrahippocampal injections were 1 nmol oligodeoxynucleotide in 0.5 µl vehicle/side, 20 min before each training session. Below left, representative paths during pretraining, the trial at the end of training, and during retention testing with the shock off 1 day after training. The shock zone is shown in red with shock on, and gray with shock off. Red circles denote where shocks are received, and gray circles where shocks would have been received if the shock were on. Right, time to first entry measure of active place avoidance memory (mean ± SEM). There is a significant interaction between the effects of genotype and treatment (scrambled, antisense) ($F_{1,39}$ = 4.14, p = 0.049, $\eta^2$ = 0.037). The individual effects of genotype and treatment are $F_{1,39}$ = 5.89, p = 0.02, $\eta^2$ = 0.053 and $F_{1,39}$ = 1.37, p = 0.25, $\eta^2$ = 0.012, respectively. Memory retention in the wild-type mice treated with PKMζ-antisense differs from the other groups (*, significant post-hoc tests; wild-types, n's = 12, PKMζ-nulls, scrambled, n = 8, antisense, 11).

The following figure supplement is available for figure 4:

**Figure supplement 1.** Fluorescence labeling of biotinylated PKMζ-antisense injected bilaterally in mouse hippocampi.

memory maintenance by intracranially injecting the PKCι/λ-inhibitor ICAP in the dorsal hippocampus only after spatial long-term memory was established, using a protocol similar to that previously used for ZIP (*Pastalkova et al., 2006*). We trained animals on active place avoidance and then 1 day later injected the PKCι/λ-inhibitor ICAP in the dorsal hippocampus, testing memory retention without shock 2 hr later (*Figure 5*). The PKCι/λ-inhibitor disrupts the retention of spatial long-term memory in PKMζ-null mice, but not in wild-type mice, as predicted by the PKMζ is compensated hypothesis.

## Increasing cognitive demand reveals place memory deficits in PKMζ-null mice

Because the data on spatial memory support the PKMζ-compensation hypothesis, we asked whether there might be differences in memory expression between wild-type mice that use PKMζ and mutant mice that rely on compensation. We examined this possibility in two ways. First, we made place avoidance more difficult to acquire by shortening the training sessions from 30 to 10 min, so that the mice require several training days for full memory expression (*Figure 6*). Clear differences in learning patterns between mutant and wild-type mice are revealed (*Figure 6A,B*, *Video 1*). In the first training trial, wild-type mice learn within minutes to move to the location opposite the shock zone. Mutant mice, although showing equivalent sensitivity to shock as wild-type mice (*Figure 6—figure supplement 1*), move to the least safe location of the rotating arena next to where it enters the shock zone. To determine whether the mutants eventually show the normal wild-type response, we extended the training over several more days. After 4 days of training, the PKMζ-null mice acquire the normal avoidance response moving to the quadrant furthest from the shock.

Like the stronger training protocol (*Figures 4*, *5*), the weaker training protocol produces measures of learning and memory in the PKMζ-null mice that are initially indistinguishable from wild-type mice (*Figure 6C*). But after 3 days, the PKMζ-null mice begin to express more errors than the wild-type mice (*Figure 6A,C*). The learning and memory deficits remain through the last training on day 5

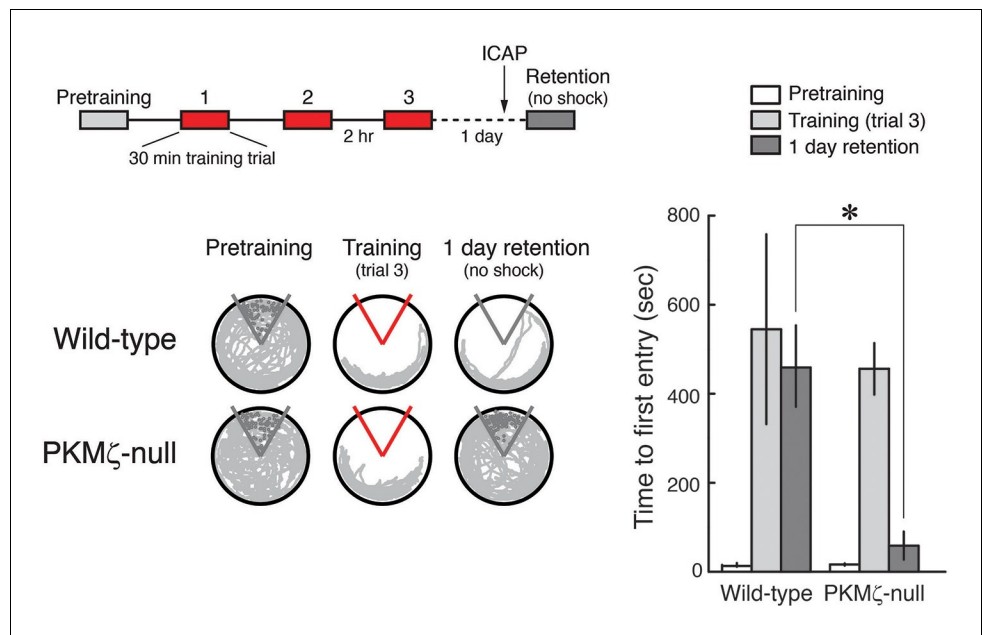

**Figure 5.** PKCι/λ inhibitor shows spatial long-term memory retention is mediated by distinct molecular mechanisms in PKMζ-null mice and wild-type mice. Insert above, schematic diagram of 1-day training protocol. Injections were 1 nmol ICAP in 0.5 μl vehicle/side, 2 hr before retention testing. Below left, representative paths during pretraining, the trial at the end of training, and during retention testing with the shock off 1 day after training. The shock zone is shown in red with shock on, and gray with shock off. Gray circles denote where shocks would have been received if the shock were on. Right, time to first entry measure of active place avoidance memory (mean ± SEM). There is a significant difference in 1-day retention between genotype ($F_{1,13}$ = 14.12, p = 0.0024, d = 1.95); wild-type, n = 8; PKMζ-null, n = 7.

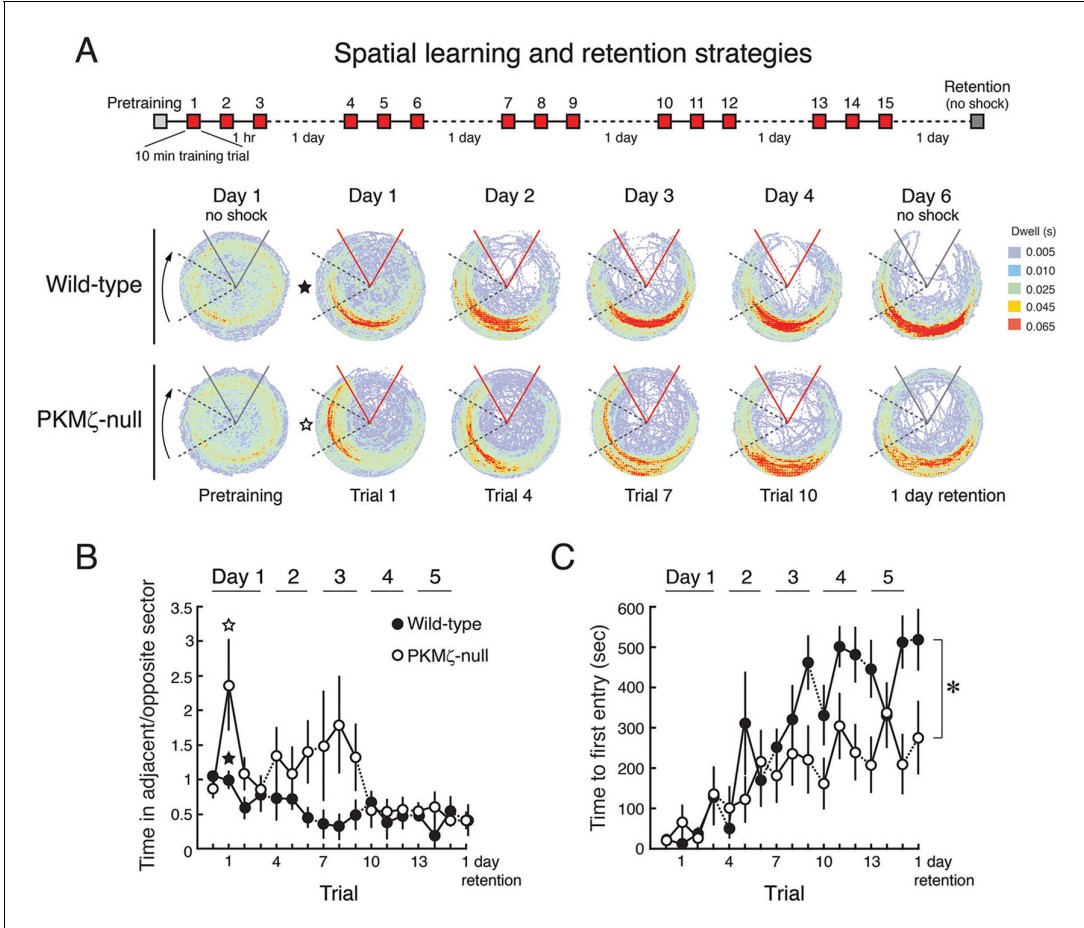

**Figure 6.** With a weaker training protocol, PKMζ-null mice show inefficient spatial learning and deficits in spatial memory. (**A**) Above, schematic diagram of 5-day training protocol. Below, color-coded time-in-location maps for wild-type and PKMζ-null mice during pretraining, beginning of training, midpoint of training, asymptote levels of performance, and 1-day memory retention without shock. In the first training trial, wild-type mice move to areas of the arena opposing the shock zone, whereas PKMζ-null mice remain in the adjacent quadrant about to enter the shock zone. Only after multiple days of training do the PKMζ-null mice show the normal avoidance behavior. N's = 10. (**B**) Plotting the ratio of time spent in the adjacent quadrant about to enter the shock zone and the safe quadrant opposite the shock zone for each trial shows PKMζ-null mice remain in the adjacent quadrant more than wild-type mice for ~9 trials over 3 days of training. Mean ± SEM; data are analyzed by genotype x trial 2-way ANOVA followed by post-hoc tests as appropriate. The PKMζ-null mice prefer being in the least efficient place for avoiding shock (genotype: $F_{1,237}$ = 16.6, p = 6.30 X $10^{-5}$, $\eta^2$ = 0.047). Effects of trial and the genotype x trial interaction are not significant. Pretraining response is denoted as trial 0. (**C**) Time to first entry into the shock zone increases to an asymptote over 3–4 days of training in both wild-type and PKMζ-null mice. Whereas the asymptote for wild-type mice is over 500 s, it is about half that for PKMζ-null mice. The main effects and interactions are all significant (genotype: $F_{1,255}$ = 22.4, p = 3.67 X $10^{-6}$, $\eta^2$ = 0.053; trial: $F_{16,255}$ = 7.3, p = 2.35 X $10^{-14}$, $\eta^2$ = 0.28; interaction: $F_{16,255}$ = 1.8, p = 0.031, $\eta^2$ = 0.068). Post-hoc tests do not distinguish PKMζ-null memory on any trial from the pretraining and first training trials when there is no avoidance memory, whereas wild-type memory is significantly better as early as day 3, trial 3, and is superior to pretraining and PKMζ-null estimates of 1-day memory from day 4 through the final retention test on day 6.

The following figure supplement is available for figure 6:

**Figure supplement 1.** Wild-type and PKMζ-null mice are indistinguishable in their motivation to escape shock during active place avoidance training.

with the two genotypes reaching distinct asymptotic levels of performance. The PKMζ-null mice also make more errors on memory testing without shock the following day.

Second, we examined spatial memory in three unconditioned novelty-preference tests that vary in their cognitive demands (*Mumby et al., 1999*), taking advantage of earlier findings showing that ZIP in hippocampus specifically disrupts information for place, but not environmental context (*Serrano et al., 2008*, *Figure 7*). As expected, when memory for discriminating the object-context associations or discriminating between a familiar and novel object-location association is tested

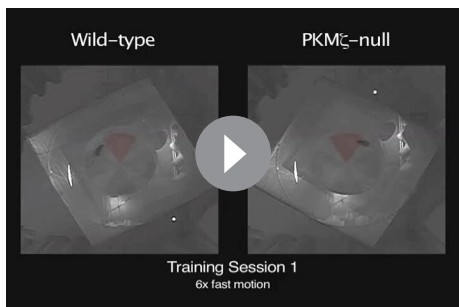

**Video 1.** Inefficient place avoidance in the PKMζ-null mouse. These videos show place avoidance behavior during the first training trial. The video on the left shows a wild-type mouse and the video on the right a PKMζ-null mouse. The wild-type mouse rapidly learns to move opposite to the shock zone to better avoid shock, whereas the mutant mouse avoids shock inefficiently by remaining in the area adjacent to the shock zone, which is the most vulnerable place.

(*Lee et al., 2013*), PKMζ-null and wild-type mice remember equally well at 24 hr. But when memory for discriminating between two familiar and novel object-location associations (object/place mismatch) is tested, wild-type mice perform well, but PKMζ-null mice show no retention of discrimination memory at 24 hr.

## Discussion

PKMζ-null mice express LTP and long-term memory, and a straightforward hypothesis to explain these results is that PKMζ is unnecessary for synaptic plasticity and memory in wild-type mice, and therefore its genetic deletion has no effect (*Lee et al., 2013*; *Volk et al., 2013*). But a second hypothesis that could account for these data is that the mechanisms of LTP and long-term memory in wild-type and PKMζ-null mice are not the same, and that PKMζ is essential for these processes in wild-type mice and compensatory mechanisms are recruited in PKMζ-null mice. We used a pharmacogenetic approach to distinguish between these hypotheses. ZIP and chelerythrine are PKMζ-inhibitors but cannot easily be used in a pharmacogenetic approach because they can block the action of both PKMζ and PKCι/λ (*Figure 1C–E*, *Figure 1—figure supplement 2*). But when ζ- and ι/λ-selective inhibitors are examined, experiments reveal a double dissociation between the mechanisms of LTP and spatial long-term memory in PKMζ-null and wild-type mice (*Figures 2*, *3*, *4*, *5*). Thus, the data indicate that the mechanisms of LTP and spatial long-term memory in wild-type and PKMζ-null mice are not the same, and the PKMζ is unnecessary hypothesis by which the function of PKMζ in wild-type mice can be inferred from the results obtained from PKMζ-null mice is erroneous.

To selectively block the action of PKMζ we took advantage of the specific nucleotide sequence of the PKMζ-mRNA translation start site to develop PKMζ-antisense oligodeoxynucleotides that suppress the activity-dependent de novo synthesis of PKMζ (*Figure 2B*). In contrast to applications of inhibitors of PKMζ's phosphotransferase activity that disrupt LTP and long-term memory after PKMζ has already been synthesized (*Ling et al., 2002*; *Serrano et al., 2005*; *Pastalkova et al., 2006*; *Shema et al., 2011*; *Cai et al., 2011*), we applied the PKMζ-antisense during the critical temporal window of new protein synthesis during late-LTP and long-term memory formation when general protein synthesis inhibitors such as anisomycin are effective (*Frey and Morris, 1997*; *Davis and Squire, 1984*), and when PKMζ is formed (*Osten et al., 1996*). Acute application of the PKMζ-antisense suppresses the new synthesis of PKMζ and blocks late-LTP without reducing basal levels of the kinase (*Figure 2B*, *Figure 2—figure supplement 2*). This suggests that the crucial pool of PKMζ protein sustaining synaptic potentiation in wild-type mice is synthesized de novo in response to tetanization, rather than through the recruitment of pre-existing PKMζ that had been synthesized before the tetanus.

To examine the mechanism maintaining late-LTP in PKMζ-null mice, we focused on the kinase most closely related to PKMζ, the other ZIP-sensitive atypical PKC, PKCι/λ. We found that whereas the increase in PKCι/λ after tetanization is transient in wild-type mice (*Osten et al., 1996*; *Kelly et al., 2007*), the increase in PKCι/λ persists in PKMζ-null mice (*Figure 3A*). In wild-type mice, PKCι/λ is important for early-LTP, and if PKCι/λ is inhibited at the time of afferent synaptic tetanization, neither early- nor late-LTP is formed (*Ren et al., 2013*). In addition, PKCι/λ-null mice, which can only be partially compensated by PKCζ, are embryonically lethal (*Seidl et al., 2013*), and therefore studies of LTP using null mutations in which both atypical PKCs are completely eliminated are not feasible. Therefore, to test for a possible role of PKCι/λ in LTP maintenance, we acutely suppressed the activity of PKCι/λ after late-LTP and spatial long-term memory had been established. We used ICAP that selectively blocks PKCι/λ and not ζ kinase activities and which was developed based upon

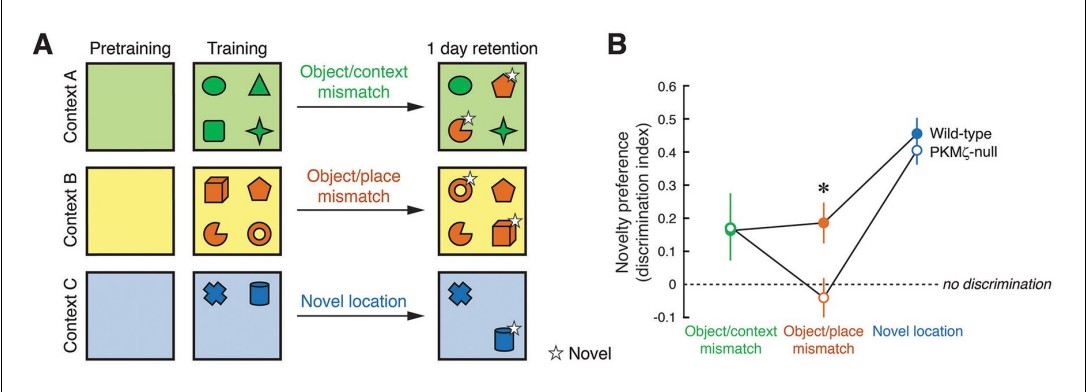

**Figure 7.** Long-term memory for where objects are encountered during exploration is tested in three unconditioned memory tests, revealing poor memory in PKMζ-null mice when the places of multiple objects are exchanged. (**A**) Mice explore distinctive boxes (contexts) during two 5 min trials per day for 3 days (4 objects) and during three 5 min trials for 1 day (2 objects). Counterbalanced object/context mismatch and object/place mismatch tests, and a novel object location test are given a day after the last exploration trial, one test per day. For clarity, only a single version of the counterbalanced tests is shown. (**B**) Both wild-type (n = 8) and PKMζ-null (n = 8) mice express similar memory discrimination when two of four objects are encountered in the incorrect context, and when one of two objects is encountered in a novel location. However, despite similar levels of investigating the objects (32–36% of the time; effect of group $F_{1,14}$ = 3.2; p = 0.09, $\eta^2$ = 0.18; effect of task $F_{2,14}$ = 0.004; p = 0.95, $\eta^2$ = 0.00049; group X task interaction $F_{1,14}$ = 0.18; p = 0.68, $\eta^2$ = 0.014), only wild-type mice express discrimination memory when the places of two of four objects are exchanged. PKMζ-null mice do not discriminate between objects that are encountered in the familiar and unfamiliar places. The interaction between genotype and memory test is significant ($F_{2,13}$ = 4.49, p = 0.03, $\eta^2$ = 0.06). The individual effects of genotype and memory test are $F_{1,14}$ = 2.23, p = 0.16, $\eta^2$ = 0.0014 and $F_{2,13}$ = 47.3, p = $10^{-6}$, $\eta^2$ = 0.025, respectively. *Significant post-hoc tests distinguish wild-type and PKMζ-null memory on the object/place mismatch test.

data from the crystal structure of the PKCι/λ catalytic domain (*Pillai et al., 2011*; *Sajan et al., 2013*; *2014*). ICAP is designed to convert in cells to a compound that targets and binds to a docking site present in the PKCι/λ catalytic domain, but not in the very closely related PKC/PKMζ catalytic domain, thus competing for binding of substrate proteins to ι/λ and not ζ (*Pillai et al., 2011*; *Sajan et al., 2013*; *2014*). The PKCι/λ-inhibitor disrupts LTP maintenance and established long-term memory in the PKMζ-null mice but not in wild-type mice (*Figures 3B,C*, *5*). But there is little information on ICAP on other kinases. Therefore, it is possible that ICAP may block the action of other molecules that may contribute to the maintenance of late-LTP and long-term memory in the PKMζ-null mice (e.g. PKCβI, which also increases in PKMζ-null mice, *Figure 3—figure supplement 1B*). Although off-target effects of ICAP are thus possible, nonetheless, the specificity of the agent in

## Box 1. Does PKMζ play a role in maintaining long-term memories for a lifetime?

The relationship between the persistent increased activity of PKMζ and the persistent changes in synaptic morphology proposed to sustain long-term memory for a lifetime is an important question for future research. The persistent action of PKMζ could function to launch or to sustain for a period the structural changes such as new synaptic connections (*Chen et al., 2014*), which once established become independent of PKMζ. Many of the remote long-term memories lasting a lifetime are thought to be stored in neocortex. Therefore, this hypothesis could be tested experimentally by examining when a long-term memory that can be erased by PKMζ inhibition in neocortex becomes resistant to this inhibition. So far, PKMζ-dependence has been tested for up to 1-3 months. In the published experiments on remote memory in neocortex, intracranial injections of ZIP disrupt 1-month-old fear memories in visual, auditory, and olfactory cortices (*Sacco and Sacchetti, 2010*), and 3-month-old conditioned taste aversion memories in insular cortex (*Shema et al., 2009*). Future studies should examine even older memories.

reversing LTP and spatial memory maintenance in PKMζ-null mice and not in wild-type mice demonstrates that the mechanisms of maintenance in the two mouse genotypes are distinct and is predicted by the PKMζ-compensation hypothesis.

The effects of ICAP on LTP maintenance only in PKMζ-null mice (*Figure 3B,C*) are thus consistent with the effects of ZIP on LTP in these mice (*Figure 1B*, *Volk et al., 2013*), because ZIP blocks the synaptic potentiation produced by both atypical PKCs, PKMζ and PKCι/λ (*Figure 1D,E*). ZIP does not block the ability of the conventional and novel PKCs to mediate synaptic potentiation, indicating selectivity of the inhibitor towards atypical PKC isoforms (*Figure 1—figure supplement 1A*). Specificity of ZIP toward atypical PKCs is further supported by evidence that ZIP's effects on LTP reversal and memory disruption in wild-type animals are completely blocked by preventing the reversal of PKMζ's action on AMPAR-trafficking, indicating the absence of additional, off-target effects of ZIP in brain slices or in vivo (*Migues et al., 2010*; *Pauli et al., 2012*; *Evuarherhe et al., 2014*). These results conflict with the non-specific toxic effects of ZIP reported on in vitro cultured neurons or at high concentrations in slices (*Sadeh et al., 2015*). We observed, however, that ZIP causes no membrane perturbation at the lower doses used to block atypical PKCs, but not conventional/novel PKCs, and to specifically reverse potentiated, but not basal synaptic transmission (*Figure 1—figure supplement 1B*, *Wang et al., 2012*).

How PKCι/λ transforms from a transiently increasing kinase in wild-type mice into a persistently increasing kinase in PKMζ-null mice is not known, but likely involves additional compensatory mechanisms that sustain PKCι/λ's synthesis or decrease its degradation during LTP. A possible role for new synthesis of PKCι/λ in PKMζ-null-LTP is consistent with data that a general protein synthesis inhibitor blocks the formation of late-LTP in PKMζ-null mice (*Volk et al., 2013*). Our results suggest the possibility that in wild-type mice PKMζ suppresses the persistent increase of PKCι/λ and that in the PKMζ-null mice without this repression the increase of PKCι/λ is sustained. PKMζ and PKCι/λ compete for protein binding partners in developing neurons, suggesting that the loss of PKMζ may allow for increased binding of PKCι/λ to these proteins that might augment its stability and function (*Parker et al., 2013*). Whereas PKMζ-null mice produce compensatory increases in PKCι/λ (*Figure 3—figure supplement 1A*), the acute suppression of PKMζ synthesis by antisense does not (*Figure 2—figure supplement 2*). Conditional PKMζ knockdown mice, which show LTP like PKMζ-null mice (*Volk et al., 2013*), have not yet been examined for compensation by PKCι/λ or other molecules. Like the antisense, shRNA knockdown of PKMζ has been found not to induce compensation by PKCι/λ, and to disrupt both late-LTP and long-term memory formation (*Dong et al., 2015*). Therefore, further work will be required to determine when and how the loss of PKMζ induces compensation by PKCι/λ or other PKCs.

Persistently increased atypical PKC activity by either PKMζ or PKCι/λ may thus be a common molecular mechanism for maintaining late-LTP and spatial long-term memory in wild-type and PKMζ-null mice, but when the cognitive demands of memory tasks increase, differences in performance emerge. When active place avoidance is made more difficult to acquire, for the first few days of conditioning wild-type mice explore the arena to find the safest location to avoid the shock, but PKMζ-null mice avoid in the least safe position adjacent to the shock zone (*Figure 6A,B*, *Video 1*). After several more days of training, the PKMζ-null mice switch to the wild-type strategy, but nonetheless perform poorly compared to wild-type mice (*Figure 6A,C*). These differences in conditioned behavior may be a sign that the PKMζ-null mice are defective in learning and memory or in integrating information about the relative safety of multiple areas of the arena. Likewise, during unreinforced novel object placement tasks, both mice with and without PKMζ remember a single object-location association discrimination, but when the number of object location-association discriminations is increased to two, wild-type mice remember well, but PKMζ-null mice express no memory for the learned information (*Figure 7*).

We speculate that some of the differences in memory expression between wild-type and PKMζ-null mice could be due to the fundamental molecular differences between PKMζ and PKCι/λ. In contrast to the second messenger-independent PKMζ, PKCι/λ can respond to second messengers that might be generated by short-term experiences (*Akimoto et al., 1994*, *Figure 3—figure supplement 2B*). Thus, in tasks with high cognitive demand, wild-type animals that express both isoforms can use PKCι/λ and PKMζ for separate functions—PKCι/λ for encoding information about short-term experiences, and PKMζ for encoding information derived from these short-term experiences to

be stored in long-term memory. But when PKCι/λ is used for both short-term memory and as a 'back-up' mechanism for long-term memory in the PKMζ-null mice, its continued responsiveness to second messengers produced by short-term experiences may interfere with its function in long-term memory. For example, during active place avoidance, when PKMζ-null mice attempt to find the safest location in an arena, information from one recently visited place might interfere with the integration of information from multiple places required for forming the optimal avoidance strategy. Likewise, in novel object placement, short-term memories of recently visited single object locations may suppress the encoding of multiple object-place associations in a single scheme. Further study of PKMζ-null mutant mice might thus reveal fundamental insights into how PKMζ encodes and stores information in long-term memory under physiological conditions in normal animals.

## Materials and methods

### Reagents

Reagents were from Sigma unless specified otherwise. The ζ-specific rabbit polyclonal antiserum (1:20,000 for immunoblots) was generated as previously described (*Hernandez et al., 2003*). Total PKCι/λ antiserum was PKCλ mouse monoclonal antibody (mAb, clone 41/PKCλ; 1:100) from BD Transduction Laboratories (San Jose, CA). The identity of PKCι/λ was confirmed with PKCι (C83H11) rabbit mAb #2998 (1:500) from Cell Signaling, Danvers, MA (*Figure 3—figure supplement 2A*) and by immunoprecipitation with PKCι/λ-specific antiserum (H-76, Santa Cruz Biotechnology, Dallas, TX) (data not shown). Phospho-atypical PKC activation loop antibody #9378 and phospho-PKC (pan, 190D10) from Cell Signaling (1:50) were raised against the same epitope, the phosphorylated form of the atypical PKC activation loop phosphorylation site. The two antisera recognize the same set of bands and gave identical results (data not shown). The eEF1A antiserum was mouse mAb, clone CBP-KK1 (1:5000), from Upstate Biotechnology (Lake Placid, NY), and the actin mouse mAb (1:5000) was from Sigma. ZIP was from Tocris Bioscience, Bristol, UK, and ICAP from United Chemical Resources, Birmingham, AL. Protein concentrations were determined by assay using bicinchoninic acid (Pierce Biotechnology, ThermoFisher Scientific, Waltham, MA) or for hippocampal extracts in reducing agents by the Bio-Rad RC-DC Protein Assay kit (Hercules, CA), using bovine serum albumin as standard.

### Genotyping and RT-PCR

Male mice from the PKMζ-null mouse line, previously described (*Lee et al., 2013*) and provided by Robert O Messing (Univ Texas at Austin, TX), were at least 4-month old at testing, and wild-type and null alleles were genotyped using primer pairs (forward: 5'-GGTATAGTAGGCAGCTATTGCG-3' and reverse: 5'-TCCTGCCTCAGCCAGAAAACAAACCACACGG-3') to identify homozygotes. All efforts were made to minimize animal suffering and to reduce the number of animals used.

For *Figure 2—figure supplement 1A,B*, mouse genomic DNA was isolated from tail biopsies using DNA Extraction Kit from Agilent Technologies, Santa Clara, CA. The final volume of PCR was 25 μl, containing 0.15 ng of genomic DNA. PCR was performed using the primer pairs (forward: 5'-GGTATAGTAGGCAGCTATTGCG-3' and reverse: 5'-TGGTGGTAAGGACAGGCTTGAGTC-3'). Amplification reactions were carried out under the following conditions: 10 mM Tris-HCl (pH 8.8), 50 mM KCL, 1.5 mM MgCl$_2$, 0.2 mM each dNTP, 0.2 μM primers, 0.06 ng/μl template, 0.04 U/μl Taq Polymerase. Genomic DNA (0.15 ng) was used in a 25 μl final volume PCR, and then 15 μl of the PCR product was loaded on a 1% agarose gel. PCR conditions were as follows: initial denaturation at 95°C for 3 min was followed by 35 cycles of denaturation at 95°C for 30 s, annealing at 64°C for 30 s, and extension at 72°C for 3 min, with final extension at 72°C for 7 min. Temperature cycling was achieved with a DNA thermal cycler (S1000 Thermal Cycler; Bio-Rad, Hercules, CA).

For real-time qRT-PCR (*Figure 2—figure supplement 1C*), total RNA was isolated from mouse hippocampi using the TRIzol reagent (Life Technologies, Carlsbad, CA) and reversed transcribed into cDNA using Superscript III (Life Technologies, Carlsbad, CA), according to manufacturer's instructions. The qPCR was performed using iQ SYBR Green Supermix Universal (Bio-Rad, Hercules, CA). Ten nanogram of cDNA was used in a 20 μl final volume PCR. Amplification was for 40 cycles with 94°C for 30 s, 60°C for 30 s, and 72°C for 30 s as cycle parameters, with a final step of 72°C for 10 min. For amplification of PKMζ cDNAs, specific primers were: forward, 5'-GGCTGCAAGAC

TTCGACCTCATC-3' and reverse, 5'-CTGGACGCCTGCTCAAACACATGT-3'. Melting curve analysis was performed to confirm the specificity of PCR reactions. Relative expression of each gene was analyzed by $\Delta\Delta C_T$ method. Data were normalized to a housekeeping gene, glyceraldehyde-3-phosphate dehydrogenase, using forward primer, 5'-TTGTGATGGGTGTGAACCACGAGA-3', and reverse primer, 5'-GAGCCCTTCCACAATGCCAAAGTT-3'. Ten microliter of the qPCR product was loaded on a 2% agarose gel to test for PKMζ mRNA expression.

## Immunoblotting

For hippocampal slice experiments, methods were adapted from those previously described (*Hernandez et al., 2003*). Briefly, slices removed from the recording chamber (see below) were immediately frozen on a glass slide on dry ice, or placed in appropriate volumes of RNA*later* solution (Ambion, ThermoFisher Scientific, Waltham, MA). The CA1 region was excised in a cold room (4°C) and homogenized in 10 µl of ice-cold modified RIPA lysis buffer, consisting of the following (in mM, unless indicated otherwise): 25 Tris-HCl (pH 7.4), 150 NaCl, 6 $MgCl_2$, 2 EDTA, 1.25% NP-40, 0.125% SDS, 0.625% Na deoxycholate, 4 p-nitrophenyl phosphate, 25 Na fluoride, 2 Na pyrophosphate, 20 dithiothreitol (DTT), 10 $\beta$-glycerophosphate, 1 µM okadaic acid, phosphatase inhibitor cocktail I & II (2% and 1%, respectively, Calbiochem), 1 phenylmethylsulfonyl fluoride, 20 µg/ml leupeptin, and 4 µg/ml aprotinin. For analysis of dorsal hippocampi and hemibrains, the tissue was dissected, snap-frozen, and stored at −80°C until lysis. Dorsal hippocampi were homogenized in 100 µl modified ice-cold RIPA buffer.

Appropriate volumes of 4X NuPage LDS Sample Buffer (Invitrogen, Carlsbad, CA) and $\beta$-mercaptoethanol were added to the homogenates, and samples were boiled for 5 min followed by SDS-PAGE. Following transfer at 4°C, nitrocellulose membranes (0.2 µm pore size) were blocked for at least 30 min at room temperature with LI-COR Odyssey Blocking Buffer (LI-COR, Lincoln, NE), then probed overnight at 4°C using primary antibodies dissolved in LI-COR Odyssey Blocking Buffer with 0.1% Tween 20 and 0.01% SDS. After washing in phosphate-buffered saline (PBS) with 0.1% Tween 20 (PBS-T; 3 washes, 5 min each), the membranes were incubated with IRDye (LI-COR) secondary antibodies. Proteins were visualized by the LI-COR Odyssey System. Densitometric analysis of the bands was performed using NIH ImageJ, and values were normalized to actin.

## Hippocampal slice preparation and recording

For LTP experiments, acute mouse hippocampal slices (450 µm) were prepared as previously described (*Serrano et al., 2005*). Hippocampi were dissected, bathed in ice-cold dissection buffer, and sliced with a McIlwain tissue slicer in a cold room (4°C). The dissection buffer contained (in mM): 125 NaCl, 2.5 KCl, 1.25 $NaH_2PO_4$, 26 $NaHCO_3$, 11 glucose, 10 $MgCl_2$, and 0.5 $CaCl_2$, and was bubbled with 95% $O_2$/5% $CO_2$ to maintain the pH at 7.4. The slices were immediately transferred into an interface recording chamber (31.5 ± 1°C) (*Serrano et al., 2005*). The recording superfusate consisted of (in mM): 118 NaCl, 3.5 KCl, 2.5 $CaCl_2$, 1.3 $MgSO_4$, 1.25 $NaH_2PO_4$, 24 $NaHCO_3$, and 15 glucose, bubbled with 95% $O_2$/5% $CO_2$, with a flow rate of 0.5 ml/min. In oligodeoxynucleotide experiments, the bath level was increased to fully submerge the slices, and the superfusate containing the oligodeoxynucleotide was recirculated (5 ml at 5 ml/min for 30 min), using a custom-made recirculation system employing piezoelectric pumps (Bartels Mikrotechnik GmbH, Dortmund, Germany). Thereafter, the bath containing the oligodeoxynucleotide was lowered again to interface level, and the flow rate was returned to 0.5 ml/min for the remainder of the experiment.

Field EPSPs were recorded with a glass extracellular recording electrode (2–5 MΩ) placed in the CA1 stratum radiatum, and concentric bipolar stimulating electrodes were placed on either side within CA3 or CA1. Hippocampal slices were excluded from study if initial analysis showed fEPSP spike threshold was <2 mV. Pathway independence was confirmed by the absence of paired-pulse facilitation between the two pathways. The high-frequency stimulation consisted of standard two 100 Hz-1 s tetanic trains, at 25% of spike threshold, spaced 20 s apart, which is optimized to produce a relatively rapid onset of protein synthesis-dependent late-LTP (*Tsokas et al., 2005*). The maximum slope of the rise of the fEPSP is analyzed on a PC using the WinLTP data acquisition program (*Anderson and Collingridge, 2007*).

For postsynaptic dialysis of PKMζ and PKCι/λ and activation of conventional/novel PKCs by bath applications of phorbol esters, hippocampal slices (400 µm) were prepared from 19- to 30-day-old

Sprague-Dawley rats, using a Vibratome tissue sectioner, as previously described (*Ling et al., 2002*). The slices were placed in an incubation chamber at 31-33°C in oxygenated (95% $O_2$, 5% $CO_2$) physiological saline consisting of (in mM): 124 NaCl, 5 KCl, 26 $NaHCO_3$, 1.6 $MgCl_2$, 4 $CaCl_2$, 10 glucose for a minimum of 1.5 hr. Single slices were then transferred to a recording chamber (1.5 ml) placed on the stage of an upright microscope (Zeiss Axioskop 2; Carl Zeiss, Oberkochen, Germany) and perfused with warm (31–33°C) saline at ~4.5 ml/min. The recording pipettes had tip resistance of 2–4 MΩ and contained (in mM): 130 Cs-MeSO₄, 10 NaCl, 2 EGTA, 10 HEPES, 1 $CaCl_2$, 2 Na-ATP, 0.5 Na-GTP. Purified PKMζ (final concentration in the pipette, 7–20 nM, 0.5–0.9 pmol·min$^{-1}$·μl$^{-1}$ phosphotransferase activity [*Ling et al., 2002*]) or PKCι/λ (final concentration, 7.4 ng/ml, 0.8 pmol·min$^{-1}$·μl$^{-1}$ [ProQinase GmbH, Breisgau, Germany]) was added to the pipette solution prior to whole-cell patch. Whole-cell recordings were obtained from visualized CA1 pyramidal cells, and synaptic events were evoked by extracellular stimulation (pulse width 0.1 ms) every 15 s with bipolar electrodes placed in stratum radiatum. The cells were held at –75 mV, and EPSC was recorded under the voltage-clamp mode with a Warner Instruments PC-501A amplifier (Hamden, CT) and filtered at 2 kHz (-3 dB, four-pole Bessel). Brief voltage steps ($-5$ mV, 5 ms) from holding potential were applied during the course of recording to monitor cell access resistance, input resistance, and capacitance. Only recordings with an initial input resistance of >100 MΩ and an initial access resistance of <10 MΩ with insignificant change (<20%) during the course of recordings were accepted for study. Signals were digitized with Digidata 1322A and acquired and analyzed with pClamp software (Molecular Devices, Sunnyvale, CA) running on a PC. The peak amplitude of EPSCs was further analyzed with Excel (Microsoft, Redmond, WA). The means ± SEMs of 1 min bins of responses were plotted in the figures.

## Antisense oligodeoxynucleotides

The sequences of the single-stranded oligodeoxynucleotides were: PKMζ-antisense, ctcTTGGGAAGGCAtgaC; scrambled, aacAATGGGTCGTCtcgG, in which the lower case bases signifies phosphorothioate linkage 5'-3'. The PKMζ-antisense sequence is complementary to the translation start site in the PKMζ mRNA and shows no significant homology to any other sequence in the GenBank database, except PKCζ mRNA. Scrambled oligodeoxynucleotide also does not match any known sequence. Both oligodeoxynucleotides are phosphorothioated on the three terminal bases at each end to protect against nuclease degradation and were reverse phase cartridge-purified (Gene Link, Hawthorne, NY) (*Garcia-Osta et al., 2006*).

For LTP experiments, oligodeoxynucleotides (20 μM) were applied to the bath after preparation of slices. A custom-made recirculation submersion system with piezoelectric pumps (Bartels Mikrotechnik GmbH, Dortmund, Germany) was used. The slices were perfused with a recirculating volume of 5 ml superfusate containing antisense- or scrambled-oligodeoxynucleotide for 1.5 hr before tetanization and for the duration of the experiment thereafter (30 min post-tetanus for immunoblots, *Figure 2B*; 3 hr for pharmacogenetic analysis, *Figure 2C*).

For spatial long-term memory experiments, we adapted the approach used in Garcia-Osta et al. (*Garcia-Osta et al., 2006*). In preparation for stereotaxic surgery to implant the injection cannula hardware, the mice were anesthetized by a mixture of dexmedetomidine (5 mg/kg i.p.) and ketamine (28 mg/kg i.p.). The animals were mounted in a Kopf stereotaxic frame (Tujunga, CA) to implant a pair of guide cannulae with the tip above the injection target in the dorsal hippocampus (AP −1.94 mm; L ±1.00 mm; DV −0.90 mm). The injection hardware was manufactured by Plastics One, Roanoke, VA (Part Numbers: C235GS-5-2.0, C235DCs-5, 303DC/1, C235IS-5; guide cannula, cannula dummy, cannula cap, injection needle, respectively). Antisedan (0.65 mg/kg i.p.) was administered to reverse the sedation at the end of surgery.

A week after surgery, the animals received active place avoidance training. Before testing the effect of the antisense injection on place avoidance, the animals received a bilateral injection of saline (1 μl/side) and were left in the home cage to habituate to the procedure. The day after the initial pretraining exposure to the place avoidance apparatus, injections were 1 nmol oligodeoxynucleotide in 0.5 μl PBS/side, 20 min before each training session. The animals were restrained, the cannula cap and dummy removed, and the injection needle inserted into the guide cannula so that it protruded from the end of the guide by 0.5 mm. The other end of the needle was connected to a 1 μl Hamilton syringe via Tygon tubing. The oligodeoxynucleotide solution was infused for 1 min, and after the infusion the needle was left in place for 5 min before removal. The animals were returned

to their home cage to recover from any acute effects of the injection and to allow diffusion of the antisense before training began. The data from 1 animal were excluded from behavioral analysis of the effects of antisense because histology revealed misplaced cannulae.

The biotinylated PKMζ-antisense (Gene Link, Hawthorne, NY) was labeled by a 5'-biotin modification with a C6 spacer. To be equivalent to the training, three biotinylated PKMζ-antisense injections with 2 hr intervals (1 nmol in 0.5 PBS μl/side) were given, and the brain fixed with 4% paraformaldehyde in PBS 50 min later. The 40 μm coronal sections were stained by immunocytochemistry using mouse anti-biotin antiserum followed by Cy3-conjugated secondary antiserum (Jackson ImmunoResearch, West Grove, PA), counterstained with DAPI, and examined by confocal microscopy.

## Kinase assay

PKMζ was recombinantly expressed and purified as previously described (*Ling et al., 2002*). PKCι/λ was purchased from ProQinase GmbH (Freiburg, Germany). The reaction mixture (50 μl final volume) contained: 50 mM Tris-HCl (pH 7.4), 10 mM $MgCl_2$, 1 mM DTT, 25 μM ε-peptide substrate (E RMRPRKRQGSVRRRV, AnaSpec, Freemont, CA), in the presence or absence of phosphatidylserine (5 μg/ml, Avanti Polar Lipids, Alabaster, AL), and PKCι/λ (184 ng, 0.2 pmol·min$^{-1}$/assay) or PKMζ (4 ng, 0.2 pmol·min$^{-1}$/assay), in the presence or absence of ZIP or chelerythrine at concentrations given in the figures. The reaction, initiated with the addition of 50 μM ATP (final concentration, ~1–3 μCi [γ-$^{32}$P]/assay), was for 30 min at 30°C, which is in the linear range for enzyme concentration (data not shown). The reaction in which the substrate LANCE Ultra ULight-PKC substrate TRF0108-D (50 nM, PerkinElmer, Waltham, MA, *Figure 1—figure supplement 2*) was substituted for ε-peptide substrate was also in the linear range for enzyme concentration (data not shown). The reaction was stopped by addition of 25 μl of 100 mM cold ATP and 100 mM EDTA, and 40 μl of the reaction mixture was spotted onto phosphocellulose paper and counted by liquid scintillation. Activity was measured as the difference between counts incorporated in the presence and absence of enzyme. Autonomous kinase activity is defined as activity in the absence of phosphatidylserine. To assay the effect of phosphoinositide-dependent kinase-1 (PDK1) phosphorylation of PKCι/λ on PKCι/λ autonomous kinase activity, PDK1 (100 ng, 8 pmol·min$^{-1}$·mg$^{-1}$; ProQinase), either active or denatured by heating to 100°C, was added to the reaction mixture. The autonomous kinase activity of PKCι/λ was then measured as described above. PKCι/λ activity in the presence of active PDK1 was normalized to that in the presence of denatured PDK1.

## Active place avoidance

A commercial computer-controlled active place avoidance system was used (Bio-Signal Group, Acton, MA). The position of the mouse on a 40 cm diameter circular arena rotating at 1 rpm was determined 30 times per second by video tracking from an overhead camera (Tracker, Bio-Signal Group). All experiments used the 'Room+Arena-' task variant that challenges the mouse on the rotating arena to avoid a shock zone that was a stationary 60° sector (*Pastalkova et al., 2006*). A constant current foot-shock (60 Hz, 500 ms) was delivered after entering the shock zone for 500 ms and was repeated each 1500 ms until the mouse left the shock zone. The arena rotation periodically transported the animal into the shock zone, forcing it to actively avoid the location of shock. The shock amplitude was 0.2 or 0.3 mA, which was determined for each animal in the first session to be the minimum that elicited flinch or escape responses.

A clear wall made from Polyethylene Terephtalate Glycol-modified (PET-G) prevented the animal from jumping off the elevated arena surface. A 5-pole shock grid was placed on the rotating arena, the centroid of the mouse was tracked by the video tracker, and the shock was scrambled across the 5-poles when the mouse entered the shock zone.

Every 33 ms, the software determined the mouse's position, whether it was in the shock zone, and whether to deliver shock. The time series of the tracked positions was analyzed offline (TrackAnalysis, Bio-Signal Group) to extract a number of end point measures. The time to first enter the shock zone estimates ability to avoid shock and was taken as an index of between-session memory. A pretraining habituation period on the apparatus equivalent in time to a training session, but without shock, was provided.

The training schedule for the pharmacogenetic analysis (*Figures 4, 5*) was as follows. The animals received three 30 min training trials, with an intertrial interval of 2 hr. Antisense or scrambled

oligodeoxynucleotide (*Figure 4*) was injected 20 min before each training trial, as described above. Retention testing was a 30 min trial without shock on the next day. ICAP (*Figure 5*) was injected 2 hr before retention testing. The mouse trajectories depict the locations that were visited during the first 10 min, the time frame during which mice that learn the avoidance tend only rarely to enter the shock zone.

The extended training schedule (*Figure 6*) was as follows. Mice were trained across a 5-day period. The animals received three 10 min training trials per day, with an intertrial interval of 1 hr. On the first day of training, animals received an additional 10 min pretraining habituation session. Retention testing was a 10 min trial without shock on the 6th day.

## Novelty-preference object spatial memory tests

Unreinforced hippocampus-dependent long-term memory was tested using adaptations of the object/context mismatch, the object/place mismatch, and the novel object location tests of novelty-preference (*Mumby et al., 1999*). The behavioral task, protocol, and analysis are described in more detail at Bio-protocol (*Lesburguères et al., 2017*). Three open plastic boxes (42 x 42 x 20 cm) were placed at the center of the experimental room. The visual appearance of each box was unique and customized with different patterns on three of the four walls to make distinct spatial contexts, denoted A, B, and C (green, yellow, blue, respectively in *Figure 7*). The fourth wall was transparent and faced south to provide orientation cues. Unique objects (toys, flasks, jars) were placed 5 cm away from the walls and were fixed to the floor. The apparatus, boxes, and objects were cleaned with 70% ethanol between subjects to eliminate odor cues. A video tracking system (Tracker, Bio-Signal Group, Acton, MA) monitored and recorded the exploratory activity of the animals for offline analysis.

Each day, the mice were placed in the experimental room 30 min before beginning the behavioral experiments. Each mouse was trained and tested in the object/context mismatch and the object/place mismatch tasks first, followed by the object-location task. Each task had three phases: pretraining, training, and the retention test. Pretraining (Day 1): the animals were allowed to explore each environment for 10 min with no objects present. Training (Days 2–4): the mice were allowed to explore a pair of environments, Contexts A and B, each during two 5 min trials/day, separated by a 1 hr intertrial interval. This allowed the mice to learn the spatial arrangement of the four objects in each box. Retention testing (Days 5 and 6): retention of spatial memory was evaluated on subsequent days by the object/context mismatch and the object/place mismatch tests. In the object/context mismatch test, two of the four objects from one context replace two of the objects in the other context, whereas in the object/place mismatch test, the positions of two objects were exchanged within one context. Each mouse was allowed to explore the altered environment for 3 min, and the time spent exploring each object was recorded. On day 5, an hour after the retention test, the mice received additional training (5 min exploration in contexts A and B with the original arrangement of objects). This additional training was intended to avoid extinction that could be induced by the retention test. The second retention test was on day 6, to test the mismatch task that was not tested on day 5. The order of the retention tests on days 5 and 6, the order of exposure to the boxes during training, the objects and the places were all counterbalanced between the animals and groups. Novel location test pretraining (Day 7): the mice explored the third environment, Context C, for 10 min. Novel location test training (Day 8): two objects were placed in this environment, and the mice explored for 5 min. During three 5 min training sessions, with 1 hr intertrial intervals, the mice could learn the locations of the pair of objects. Novel location retention test (Day 9): one object was relocated to a novel unoccupied place, and the time spent exploring each object was recorded. The relocated object, the location, and the specific environment were counterbalanced between animals and genotypes.

Offline analysis of the video identified exploration when the mouse's nose was <2 cm away and oriented toward the object. Memory performance was quantified using a discrimination index calculated as the absolute difference in time spent exploring the changed (i.e. incorrect, misplaced, or relocated) and the unchanged objects divided by the total time spent exploring all the objects. Good memory retention corresponds to a positive discrimination index, which reflects that the animal spent more time exploring the incorrect (object/context mismatch), displaced (object/place mismatch), or relocated (object-location) objects compared to the objects that were not changed.

## Statistics

Sample sizes vary for the different experimental approaches (biochemistry, in vitro intracellular current and extracellular field potential physiology, and behavior). The PKMζ is unnecessary and the PKMζ is compensated hypotheses predict all-or-none effects in the experiments, and this provided a basis for sample size estimates. Power analyses were performed using G*Power Version 3.0.6 with $\alpha = 0.05$ and $\beta = 0.8$ and large effect sizes of 1.5–2.0. The effect size estimates were based on prior studies that demonstrated essentially all-or-none effects of PKMζ inhibition on the biochemical, physiological, and behavioral assays used here (*Sacktor et al., 1993*; *Osten et al., 1996*; *Ling et al., 2002*; *Kelly et al., 2007*; *Pastalkova et al., 2006*). Two-population Student's t tests were performed to compare protein levels in the PKMζ-null and wild-type mice. For LTP experiments the responses to test stimuli were averaged across 5 min for statistical comparisons. Paired Student's t tests were used to compare the change in the potentiated response at time points at the beginning and end of drug application. Multi-factor comparisons were performed using ANOVA with repeated measures, as appropriate. The degrees of freedom for the critical t values of the t tests and the F values of the ANOVAs are reported as subscripts. Post-hoc multiple comparisons were performed by Newman-Keuls tests as appropriate. Statistical significance was accepted at $p < 0.05$. Effect sizes for binary comparisons and one-way ANOVAs are reported as Cohen's d and as $\eta^2$ for two-way ANOVA effects.

## Acknowledgements

We are grateful to the late Bob Muller for his comments and dedicate this paper to his memory. We thank Juan Marcos Alarcon, Jeremy Barry, John Kubie, Terje Lømo, Alice Pavlowsky, Wayne Sossin, and Robert KS Wong for helpful comments and discussions. We thank Randy Andronica for technical assistance and Rose Sacktor for help with illustrations. PT is an Alexander S Onassis Public Benefit Foundation Scholar.

## Additional information

### Funding

| Funder | Grant reference number | Author |
| --- | --- | --- |
| National Institute of Mental Health | R37 MH057068 | Todd Charlton Sacktor |
| National Institute of Mental Health | R01 MH53576 | Todd Charlton Sacktor |
| National Institute on Drug Abuse | R01 DA034970 | Harel Z Shouval<br>Todd Charlton Sacktor |
| Lightfighter Trust | | Todd Charlton Sacktor |
| National Institute of Mental Health | R01 MH084038 | André Antonio Fenton |
| National Institute of Mental Health | R01 MH099128 | André Antonio Fenton |
| National Institute on Aging | R01 AG043688 | André Antonio Fenton |
| National Science Foundation | IOS-1146822 | André Antonio Fenton |

The funders had no role in study design, data collection and interpretation, or the decision to submit the work for publication.

### Author contributions

PT, CH, YY, EL, AT, DJ, BRH, LP, AIH, AAF, TCS, Conception and design, Acquisition of data, Analysis and interpretation of data, Drafting or revising the article, Contributed unpublished essential data or reagents; EJCW, Acquisition of data, Analysis and interpretation of data, Drafting or revising the article, Contributed unpublished essential data or reagents; BR, RVF, MPS, PJB, JEC, HZS, Conception and design, Drafting or revising the article, Contributed unpublished essential data or reagents

## Author ORCIDs

Panayiotis Tsokas, http://orcid.org/0000-0002-1679-8980
Changchi Hsieh, http://orcid.org/0000-0003-0488-1565
André Antonio Fenton, http://orcid.org/0000-0002-5063-1156
Todd Charlton Sacktor, http://orcid.org/0000-0002-8625-8701

## Ethics

Animal experimentation: This study was performed in strict accordance with the recommendations in the Guide for the Care and Use of Laboratory Animals of the National Institutes of Health. All of the animals were handled according to approved institutional animal care and use committee (IACUC) protocols (#11-10274, #12-10298, #13-10363 of SUNY Downstate Medical Center or #15-1459 of New York University). The protocols were approved by the Institutional Animal Care and Use Committee of SUNY Downstate Medical Center (Animal Welfare Assurance Number: A3260-01) and New York University (Animal Welfare Assurance Number: A3317-01). All surgery was performed under either ketamine and dexmedetomidine, isoflurane, or sodium pentobarbital anesthesia, and every effort was made to minimize suffering.

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
