## [Decision Letter]

Thank you for submitting your article "Compensation for PKMζ in LTP and spatial long-term memory in mutant mice" for consideration by *eLife*. Your article has been reviewed by two peer reviewers, including Richard Morris, and the evaluation has been overseen by Gary Westbrook as the Senior Editor.

The reviewers have discussed the reviews with one another and the Reviewing Editor has drafted this decision to help you prepare a revised submission. Both reviewers were very supportive of the manuscript and requested only very minor revisions and clarifications. Specifically, please clarify in the revised manuscript your view of the issue raised by Reviewer 1 as their "main concern" and address the statistical issue raised by Reviewer 2.

Reviewer #1:

I am so very pleased to have read this manuscript submitted to *eLife* as I have been following the PKMζ story closely, I am not alone in doing so, and I feel sure that a large number of people will wish to read and cite a published version of this manuscript. This is – surely – important work. My reading of the manuscript reveals some questions, which I will come to, but my sense is that the key experiments have been conducted to test the idea that PKMζ is compensated by another atypical PKC in PKMζ null mice, and that this mediates late-LTP and long term memory in such genetically modified mice. The work involves the use of a non-discriminating drug (ZIP), specific antisense molecules, and another compound that targets the compensating PKCι. And the manuscript is beautifully written and accompanied by excellent figures.

My main concern

Over 20 years, Dr Sacktor has been building a story about identifying the likely molecular mechanism of long-term memory. The driving concept behind this is stated clearly in the first paragraph of the Introduction in which it is asserted that the hypothesis of a molecule implicated in maintenance has specific experimental predictions. I fear that the senior author's view on this is *not* shared by others as a third possibility is not considered – namely that a molecule implicated in maintenance, while to be sure not contributing to initial memory formation, might not be sustained throughout the lifetime of a memory. To the contrary, it may trigger structural changes mediated by other molecules, and then depart the scene and no longer be involved. These structural changes would then be recycled during protein turnover in some way (but that is a separate albeit related problem in memory). An analogy might be helpful here. Consider a space rocket already in space that is circling the earth and is to be sent to the moon. For a brief period, the thrusters are activated and the rocket escapes earth's gravity and speeds up. It is on its way to the moon. The engines are stopped and the rocket keeps going. Should we look for the 'molecules' that sustain its motion towards the moon, akin to maintaining a memory as in Sacktor's argument? Or do we recognise that Newton's Laws of Motion distinguish acceleration and velocity and recognise that nothing is needed to sustain velocity in space? The molecules that make memory retention possible could, like the engines on a rocket, be activated only briefly.

The key findings of the manuscript

In any event, while not sharing exactly Sacktor's perspective, I am wholly persuaded by the impressive data that his group have secured over the years to the effect that PKMζ is an important memory molecule. Unfortunately, the story was seriously dented by the publication in Nature in 2013 of two papers establishing normal LTP and normal long-term memory (LTM) in PKMζ knockout mice. These papers were widely discussed and the general 'backchat' is that they had demolished Sacktor's story. But 'backchat' is not always to be relied upon – as this manuscript now so beautifully demonstrates.

Sacktor now comes back in this manuscript with convincing evidence that a different atypical PKC, called PKCι/λ (I believe), is upregulated in the absence of PKMζ and may take over some of its functions. In PKMζ null mice, ZIP continues to be effective in reducing late-LTP but it may be that this happens because ZIP, while previously thought to be specific to PKMζ also has an effect on PKCι/λ.

The way this new body of work takes us further is by first developing, and then using, a very specific antisense molecule targeting the translation start site of PKMζ. The prediction is that this should successfully affect late-LTP in wild-type mice but not in PKMζ null mice. This prediction is upheld (Figure 2) and validated through appropriate biochemical experiments (Figure 2).

The next step was to monitor the temporal expression of PKCι/λ after LTP induction. Nicely, it increases transiently in wild-type mice (in which the upregulation of PKMζ is of course doing the work), but it is increased for very much longer (>3 hrs) in PKMζ knockouts (Figure 3). The group went on to test a molecule ICAP that affects PKCι/λ selectively and establish that its application reverses late-LTP in PKMζ null mice but not in wild-type mice (Figure 3). This leaves the issue of why PKCι/λ is upregulated for longer, a point made in Discussion with sensible remarks accompanying.

Behavioural studies then become the focus with the demonstration using a place avoidance task first developed in the Czech Republic that PKMζ-antisense successfully disrupts LTM in wild-type mice but not in PKMζ null mice. Conversely, ICAP blocks LTM in PKMζ null mice in which such memory is presumed to be mediated, in part, by the compensatory PKCι/λ molecule. Last, some data are presented showing in behavioural studies that the compensation is not 'perfect' and that subtle differences in strategy can be seen between wild-type and PKMζ null mice in their initial learning of the place avoidance task when somewhat less aggressive training is deployed.

I have described the results in some detail as they seem to me to constitute a serious test of two conflicting hypotheses – as presented at the outset of manuscript – to the effect that either (1) PKMζ plays no role in long-term maintenance (Huganir), or (2) that it does so in wild-type mice, but its absence is effectively compensated in PKMζ gene knock-out mice.

Conclusion

I strongly recommend publication – essentially as the manuscript is written. I wondered if Sacktor should add that one reason for compensation may be because PKMζ is so important for memory that an organism cannot afford to be without the necessary mechanism if the gene were to malfunction – but perhaps that is too speculative.

Reviewer #2:

The study by Tsokas et al. is a remarkably thorough and interesting test of the hypothesis that PKMζ, an atypical protein kinase C isoform, mediates protein synthesis-dependent late-phase LTP in wild type mice. The authors are responding to a finding that mice, in which PKMζ is knocked out by homologous recombination, still show late LTP and that LTP is still inhibited by an inhibitor of PKMζ termed ZIP. The authors correctly state that the latter findings support either of two hypotheses: 1. PKMζ is unnecessary for long term memory and late-phase LTP; or 2. PKMζ is essential for these processes in wild-type mice, and compensatory mechanisms emerge in the PKMζ mutant to substitute for PKMζ. They systematically show that in the hippocampus, increased synthesis of PKCι/λ compensates for the lack of PKMζ in the mutant. PKCι/λ, like PKMζ, is inhibited by ZIP, which explains the earlier results in the PKMζ KO mutant.

The authors show that bath application of phosphorothioated antisense DNA oligonucleotides that are specific for the mRNA encoding PKMζ, specifically block new synthesis of PKMζ during late-phase LTP and also block formation of late-LTP in slices from wild type mice. Scrambled antisense oligonucleotides are the control. In contrast, the PKMζ specific oligonucleotides do not block late phase LTP in slices from the PKMζ KO mutant. This experiment demonstrates that a compensatory mechanism not involving PKMζ has arisen in the mutant mice. I note that in cultured neurons, one would expect the experimenters to show rescue of late-LTP by exogenously expressed PKMζ in conjunction with endogenous anti-sense knock-down. I do not think this particular control is necessary here because it would be very difficult to carry out cleanly in slices and animals; and later experiments by the authors make it clear that the compensatory process involves higher synthesis of PKCι/λ, which they show is not blocked by the anti-sense DNA against PKMζ.

The authors then show that an inhibitor, termed ICAP, that is specific for inhibition of PKCι/λ inhibits late-LTP in PKMζ KO mutants, but does not inhibit late-LTP in wild type mice. They further show that new synthesis of PKCι/λ increases after induction of LTP in PKMζ KO mice, but does not increase in wild-type mice. These experiments, together with the catalytic similarity between PKCι/λ and PKMζ, make a strong case that PKCι/λ compensates for the loss of PKMζ in the KO mice, during late-LTP.

In a supplementary figure, the authors show the very interesting finding that an increased level of basal PKCι/λ is measurable in the hippocampus of PKMζ KO mice, but not in the contralatera hemi-brain. Thus, the role of PKCι/λ may be more prominent in hippocampus than in cortex. This finding explains the earlier report that PKCι/λ is not increased in expression in KO brains.

The authors then show that PKMζ antisense blocks retention of a spatial memory, shock-avoidance task in wild-type mice, but not in PKMζ KO mice. Thus, the compensation extends to behavioral measures. They further show that the retention is blocked in PKMζ KO mice by ICAP, but is not blocked by ICAP in wild-type mice. Finally, the authors show that certain forms of memory are not as efficient in the PKMζ KO mice as in wild-type; thus, demonstrating that the compensation by PKCι/λ is only partially effective.

My only critical comments are minor, involving clarity of the statistical measures, and are shown below.

To my knowledge, this is the most complete demonstration of a compensatory mechanism in synapses following KO of a critical component. In addition to proving that PKMζ indeed plays a necessary role in maintenance of late phase LTP, the study provides interesting insight into the precision of such mechanisms underlying behavioral memory. The authors are to be congratulated.

The statistical analyses of the authors' experiments are extensive; but are only partially explained in the Methods section. The authors do not indicate the meaning of the F measures or the t measures in the Methods section. This author was able to decipher the meaning of the t measure (with its subscript) in the students t-test. However, the meaning of the subscripts for the F measures was not clear to me. I understand that F is a measure of the ratios of variance of two means. However, I don't understand the source of the subscripts for F in the authors figure legends. The authors should give a bit more explanation of their t measure and their F measures in the Methods section.

---

## [Author Response]

The reviewers have discussed the reviews with one another and the Reviewing Editor has drafted this decision to help you prepare a revised submission. Both reviewers were very supportive of the manuscript and requested only very minor revisions and clarifications. Specifically, please clarify in the revised manuscript your view of the issue raised by Reviewer 1 as their "main concern" and address the statistical issue raised by Reviewer 2. Reviewer #1:

*My main concern Over 20 years, Dr Sacktor has been building a story about identifying the likely molecular mechanism of long-term memory. The driving concept behind this is stated clearly in the first paragraph of the Introduction in which it is asserted that the hypothesis of a molecule implicated in maintenance has specific experimental predictions. I fear that the senior author's view on this is not shared by others as a third possibility is not considered – namely that a molecule implicated in maintenance, while to be sure not contributing to initial memory formation, might not be sustained throughout the lifetime of a memory. To the contrary, it may trigger structural changes mediated by other molecules, and then depart the scene and no longer be involved. These structural changes would then be recycled during protein turnover in some way (but that is a separate albeit related problem in memory). An analogy might be helpful here. Consider a space rocket already in space that is circling the earth and is to be sent to the moon. For a brief period, the thrusters are activated and the rocket escapes earth's gravity and speeds up. It is on its way to the moon. The engines are stopped and the rocket keeps going. Should we look for the 'molecules' that sustain its motion towards the moon, akin to maintaining a memory as in Sacktor's argument? Or do we recognise that Newton's Laws of Motion distinguish acceleration and velocity and recognise that nothing is needed to sustain velocity in space? The molecules that make memory retention possible could, like the engines on a rocket, be activated only briefly.* The relationship between the persistent increased activity of PKMζ and the persistent changes in synaptic morphology proposed to sustain long-term memory for a lifetime is an important question for future research. The persistent action of PKMζ could act to launch or to sustain for a period the structural changes such as new synaptic connections (Chen et al., *eLife*, 3, e03896, 2014), which once established become independent of PKMζ. Many of the remote long-term memories lasting a lifetime are thought to be stored in neocortex. Therefore, this hypothesis could be tested experimentally by examining when a long-term memory that can be erased by injections of ZIP in neocortex becomes resistant to PKMζ inhibition. So far, PKMζ-dependence has been tested for up to 3 months. In the published experiments on remote memory in neocortex, intracranial injections of ZIP disrupt 1-month-old fear memories in visual, auditory, and olfactory cortices (Sacco and Sacchetti, 329, 649-56, 2010), and 3-month-old conditioned taste aversion memories in insular cortex (Shema et al., 16, 122-8, 2009). Future studies should examine even older memories.

The discussion of this key question on memory is highlighted in [Box B1].

Reviewer #2:

My only critical comments are minor, involving clarity of the statistical measures, and are shown below.

*The statistical analyses of the authors' experiments are extensive; but are only partially explained in the Methods section. The authors do not indicate the meaning of the F measures or the t measures in the Methods section. This author was able to decipher the meaning of the t measure (with its subscript) in the students t-test. However, the meaning of the subscripts for the F measures was not clear to me. I understand that F is a measure of the ratios of variance of two means. However, I don't understand the source of the subscripts for F in the authors figure legends. The authors should give a bit more explanation of their t measure and their F measures in the Methods section.*

As requested, we now include in the Methods that the degrees of freedom for the critical t values of the t tests and the F values of the ANOVAs are reported as subscripts (in the subsection “Statistics”).